# Can Computational Reducibility Lead to Transferable Models for Graph Combinatorial Optimization?

Semih Cantürk [* 1 2]   Thomas Sabourin [* 3 2]   Frederik Wenkel [† 4]   Michael Perlmutter [5]   Guy Wolf [3 2]

## Abstract

A key challenge in deriving unified neural solvers for combinatorial optimization (CO) is efficient generalization of models between a given set of tasks to new tasks not used during the initial training process. To address it, we first establish a new GNN encoder, which uses a GCON module as a form of expressive message passing together with energy-based unsupervised loss functions. This model achieves highly competitive performance across multiple CO tasks when trained individually on each task. We then leverage knowledge from the computational reducibility literature to propose pretraining and fine-tuning strategies that transfer effectively (a) between MVC, MIS and MaxClique, and (b) in a multi-task learning setting that additionally incorporates MaxCut, MDS and graph coloring. Additionally, in a leave-one-out, multi-task learning setting, we observe that pretraining on all but one task almost always leads to faster convergence on the remaining task when fine-tuning while avoiding negative transfer. Our findings indicate that learning common representations across multiple graph CO problems is viable through the use of expressive message passing coupled with pretraining strategies that are informed by the polynomial reduction literature, thereby taking an important step towards enabling the development of foundational models for neural CO.

---
[*]Equal contribution [†]Work done while affiliated with [1, 2]. [1]Dept. of Computer Science and Operations Research, Université de Montréal [2]Mila – Quebec AI Institute [3]Dept. of Mathematics & Statistics, Université de Montréal [4]Valence Labs [5]Dept. of Mathematics, Boise State University. Correspondence to: Semih Cantürk <semih.canturk@umontreal.ca>, Thomas Sabourin <thomas.sabourin@umontreal.ca>.

*Proceedings of the 43rd International Conference on Machine Learning*, Seoul, South Korea. PMLR 306, 2026. Copyright 2026 by the author(s).

## 1. Introduction

The ability to transfer performance from one task to another is a crucial challenge in the development of modern AI since it eliminates the need to train a model from scratch each time that a new task is encountered. Instead, we seek to train a model to perform well on a representative collection of problems in such a way that it may be easily adapted to each new task in a relatively lightweight manner. Common techniques for this include partial fine-tuning as well as attaching relatively simple downstream task heads to an otherwise frozen architecture (Oquab et al., 2014; Zeiler & Fergus, 2014; Devlin et al., 2019; You et al., 2021; Kaur et al., 2021). For example, in medical image analysis, pretrained vision models have been shown to provide meaningful features, significantly improving conventional ML efficacy (Kaur & Mahajan, 2025), and a promising initialization for further fine-tuning under data scarcity conditions (Eliwa, 2025).

We note that while transferability has only recently emerged to the forefront of modern ML/AI, it has been a cornerstone of theoretical CS for many decades. Indeed, the study of computability and complexity relies on a hierarchical structure of equivalence classes defined by appropriate reductions between problems (Cormen et al., 2022; Papadimitriou & Steiglitz, 1998). Perhaps the most prominent examples are the classes P and NP, containing problems that are solvable in deterministic vs. non-deterministic polynomial time (correspondingly). To show that a problem is in P, one has to either find a direct algorithm that solves it, or – perhaps more often – an indirect solution via a polynomial *reduction* to another problem that is known to be in P. This reduction would translate the inputs to the target problem, and then translate the solution from it back to that of source problem. As long as both translation steps can be done in polynomial time, their cascade with the polynomial algorithm for the target problem would also result in a polynomial algorithm. Similarly, membership in NP can be established either by verifying a hypothesized solution in polynomial time, either directly, or indirectly via reduction to a known NP problem (i.e., by translating the candidate solution to test on it). In particular, the study of such reductions gives rise to the notion of NP-complete (or NP-hard) problems – those to which every problem in NP can be (polynomially) re-

duced. Indeed, these problems form an upper bound on NP complexity, in the sense that if any of them is solvable in polynomial time, it would immediately enable polynomially solving any problem in NP via a cascade of reductions.

We note that while the discussion above focused on polynomial complexity, numerous other complexity classes have been extensively studied, such as poly-logarithmic, linear, or randomized-polynomial time, as well as ones focusing on other algorithmic aspects, such as parallelization capabilities and space requirements. A common theme in each of these is the reliance on complexity-bounded reductions, yielding the notion of completeness (or hardness) of problem classes. Further, in these studies, typically only a handful of core problems that are directly solved via a handcrafted algorithm, whereas the other problems are indirectly solved by reducibility. We refer the reader to Cormen et al. (2022); Papadimitriou & Steiglitz (1998) for further details and extensive background on these topics. Intuitively, one could argue that the aim of foundation models in deep learning is similar, at least in spirit, to these studies, in that they aim to establish a core set of tasks (with associated loss terms and curated data) that are sufficient to elicit emergent behavior, by which entire families of problems (or tasks) will be easily, if not trivially, approached with relatively simple adaptation of the core foundation model.

In this work, we investigate whether the notion of efficient (e.g., complexity-bounded) reducibility can inspire, or potentially inform, decisions regarding transferability. This is particularly relevant in combinatorial optimization problems on graphs, which have been of high interest in recent graph neural network studies, as well as in theoretical computer science, motivated in part by applications in, e.g., logistics (Bao et al., 2018), health care (Zhong & Tang, 2021), and scientific discovery (Naseri & Koffas, 2020).

Notably, most graph CO problems are NP-hard and feature an exponential search space, which makes their direct solution computationally prohibitive. Nevertheless, numerous recent work have produced promising results for generating approximate solutions for individual CO tasks, often using networks that are trained by task-specific loss functions (Lucas, 2014; Karalias & Loukas, 2020; Min et al., 2022; Zhang et al., 2023). Here, we seek to build on their work towards a unified model that is able to solve multiple CO tasks simultaneously, utilizing connections between transferability and reducibility. In particular, we aim to replace traditional reductions (driven by complexity bounds) with efficient transfer learning mechanisms suitable for integration in deep learning paradigms, and establish empirically their cross-task generalization feasibility. In doing so, we aim to provide an important stepping stone towards the development of foundation models that form a universal neural solver for CO problems.

To this end, we first establish, in Sec. 3, new baselines for individual CO tasks based on the Graph Combinatorial Optimization Network (GCON) network introduced by Wenkel et al. (2025) and show that this method matches the performance of most current methods, and occasionally even establishes a new state-of-the-art (albeit not our primary focus here). Namely, we focus on the following six problems, described in further detail in Sec. 3.3: Maximum Independent Set (MIS), Minimum Dominating Set (MDS), Minimum Vertex Cover (MVC), Max Clique, Max Cut, and Graph Coloring. We then turn our attention to transferability, first studying in Sec. 4 pairwise transferability between three of these tasks. There, we explore the relation between reducibility and transferability, while also elucidating the challenges in translating theoretical knowledge to practical application in deep learning settings. Next, in Sec. 5 we shift towards a more realistic setting, addressing a primary prerequisite for the development of transferable foundation models, which are based on forming a unified trunk pre-trained via multi-task learning, and that can generalize to new tasks with relatively light-weight adaptation.

Our results indicate that indeed some connections arise between the notions of reducibility and transferability, although these connections are not trivial and further work is needed to fully elucidate them. Nevertheless, we see our contributions as providing a stepping stone for further studies on this topic, and a promising starting point for future work on universal neural CO solvers.

## 2. Related Work

Several prominent lines of research have emerged as part of the recent interest in graph CO problems. We focus here on a brief summary of the ones most relevant to this work. A more extensive discussion is provided in Appendix B.

In particular, we focus here on unsupervised approaches, e.g., Karalias & Loukas (2020), which proposed a surrogate loss function for the maximum clique and graph partitioning problems, together with sequential decoding of probabilistic predictions to obtain valid solutions. Min et al. (2022) paired this framework with a GNN derived from the geometric scattering transform (Gao et al., 2019) to obtain further competitive results on approximate max clique, while Min et al. (2023) used a similar setup for the traveling salesman problem. A recent subset of unsupervised GNN-based solvers use *physics-inspired* loss functions based on Lucas (2014), which showed that many CO problems map to the Ising model, and minimizing the corresponding Hamiltonian representing its energy landscape is equivalent to solving them. Schuetz et al. (2022a); Krylova & Phillipson (2025) tackle max cut and MIS using this recipe while Schuetz et al. (2022b) extend it to graph coloring. More recently, Sanokowski et al. (2024) used task-specific Hamiltonians to

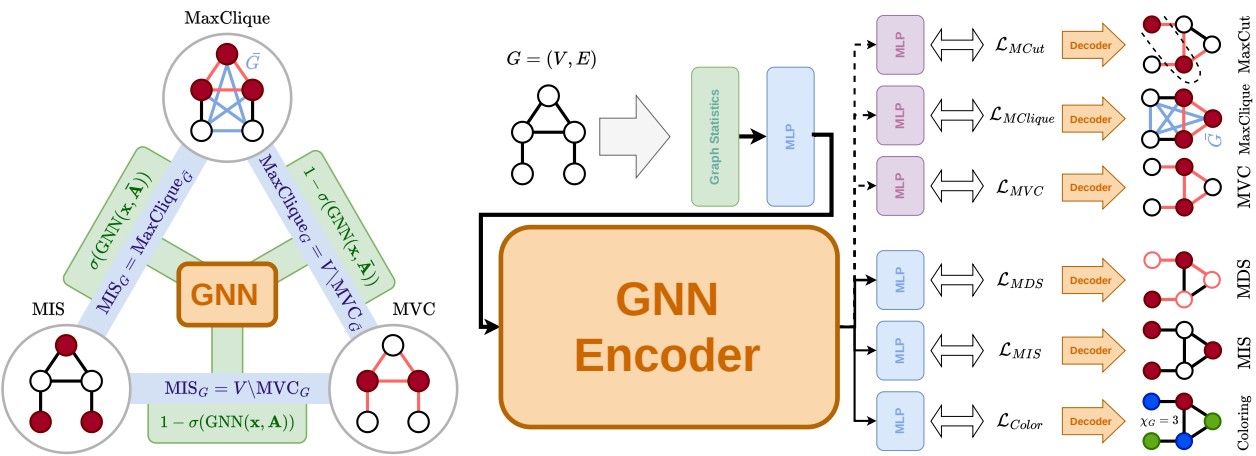

*Figure 1.* (Left) Reduction between tasks for pairwise transferability section: MIS/MVC are complements, but MaxClique is based on an auxiliary graph. (Right) Multi-task learning / fine-tuning architecture (pretraining set in green, fine-tuning set in purple).

train a diffusion model and attain competitive results across multiple tasks.

**Multi-Task neural CO.** Most attempts to develop a unified model for CO problems are based on supervised or reinforcement learning. Examples include GOAL (Drakulic et al., 2024) based on imitating expert trajectories, UniCO (Zong et al., 2025) based on a Markov Decision Process, and the RL models in Li et al. (2025); Wang et al. (2025) based on gradient homogenization and a multi-armed bandits respectively. Another approach aims to obtain a generic representation via SAT bipartite graphs (Guo et al., 2025; Zeng et al., 2023) or syntax trees capturing problem constraints (Boisvert et al., 2024). Unlike these methods, we rely on unsupervised learning with task-specific loss functions, which avoids problems arising from the large size of the state space in RL. Moreover, we avoid transforming the graphs to a common SAT or abstract representation but rather work on the original graph structure by using a GCON backbone that leverages a complex filter bank and localized attention mechanisms to learn approximate solutions to CO problems (Wenkel et al., 2025). Finally, unlike most of these methods, which do not justify their selection of pretraining tasks, we aim to use known polynomial reductions to show similar problems transfer effectively between each other, thus guiding our pretraining tasks selection. See Appendix B for further discussion.

**Polynomial reductions for CO problems.** Combinatorial optimization problems have been extensively studied in the computer science literature, e.g., with Karp's 21 NP-complete problems (Karp, 1972). Many polynomial reductions are well known between these problems, such as the equivalence relations shown in Garey & Johnson (1979) between the maximum clique (MaxClique), maximum independent set (MIS) and minimum vertex cover (MVC) problems (see Lemma B.1 in Appendix B). Moreover, Filar et al. (2019) demonstrate the existence of a kernel subset of Karp's 21 problems, which span the rest of the problems via the notion of linear orbit, defined via linear reducibility. In particular, their work implies that MaxClique, MaxCut, Node Cover (MVC and therefore MIS), and Set Cover (of which MDS is a special case) reside in the linear orbit of 0-1 Programming. Note that, conversely, problems like Chromatic Number ($K$-coloring) and the Hamiltonian Cycle Problem appear to occupy different linear orbits. A formal discussion and further details are provided in Appendix B.

## 3. Methodology

*Table 1.* Energy-based loss functions for CO problems considered. Table extended from Sanokowski et al. (2024).

| CO PROBLEM | OBJECTIVE: $\min_{X \in \{0,1\}^N} H(X)$ |
|---|---|
| MIS | $H(X) = -A \sum_{i=1}^N X_i + B \sum_{(i,j) \in \mathcal{E}} X_i \cdot X_j$ |
| MDS | $H(X) = A \sum_{i=1}^N X_i + B \sum_{i=1}^N (1 - X_i) \prod_{j \in \mathcal{N}(j)} (1 - X_j)$ |
| MAXCLIQUE | $H(X) = -A \sum_{i=1}^N X_i + B \sum_{(i,j) \notin \mathcal{E}} X_i \cdot X_j$ |
| MAXCUT | $H(\sigma) = -\sum_{(i,j) \in \mathcal{E}} \frac{1 - \sigma_i \sigma_j}{2}$ WHERE $\sigma_i = 2X_i - 1$ |
| MVC | $H(X) = A \sum_{i=1}^N X_i + B \sum_{(i,j) \in \mathcal{E}} (1 - X_i) \cdot (1 - X_j)$ |
| $K$-COLORING | $H(X) = \sum_{i=1}^N (1 - \sum_{k=1}^K X_{i,k})^2 + \sum_{(i,j) \in \mathcal{E}} \sum_{k=1}^K X_{i,k} X_{j,k}$ |

We frame each CO task as an unsupervised learning problem, and assume we do not have access to the set of interest or even the size of this set. Throughout, we assume undirected and unweighted graphs $G := (V, E)$, $V = \{v_1, \ldots, v_N\}$, though much of our work can easily be extended to weighted graphs. We denote the complement of $G$ as $\bar{G} := (V, \bar{E})$.

When training models from scratch, our pipeline largely follows Wenkel et al. (2025) and Karalias & Loukas (2020).

We first train a GNN encoder, which is trained by a task-specific, unsupervised loss function to generate a vector $\mathbf{p}$ whose $i$-th entry $\mathbf{p}_i = \mathbf{p}[v_i]$ is interpreted as the probability that $v_i$ is in the set of interest. That is, the $\mathbf{p}_i$ may be thought of a soft proxies for the indicator variables $X_i \in \{0, 1\}$ which indicate whether $v_i$ is a member of this set. We then use a sequential, rules-based decoder that strictly enforces the problem constraints to arrive at the final set of interest. We discuss the individual elements of our pipeline below.

We note that throughout this work, we will frequently use the terms reducibility and transferability. By reducibility, we refer to the existence of a polynomial-time reduction between computational problems in the standard sense of theoretical computer science, as used in complexity theory to relate decision or optimization problems. By transferability, we refer to the machine learning paradigm in which knowledge acquired from source tasks is leveraged to improve performance on a related target task. This can be achieved through various mechanisms, including partial parameter reuse (e.g., adapting only task-specific components such as output heads) or full fine-tuning of all model parameters from a pretrained initialization.

### 3.1. GNN Encoder

Our GNN encoder relies on the Graph Combinatorial Optimization Network (GCON) architecture introduced in Wenkel et al. (2025). Unlike local message-passing GNNs such as GCN, GIN or GAT which effectively perform low-pass filtering over the graph, GCON uses a rich bank of multi-scale wavelet filters inspired by the geometric scattering transform (Gao et al., 2019; Gama et al., 2018; Zou & Lerman, 2020; Perlmutter et al., 2023; Chew et al., 2024). Wenkel et al. (2025) demonstrate that such networks can learn rich node representations, and avoid typical information bottlenecks of localized message-passing.

We follow a a common architectural design of a linear layer before and after the GNN layers, as well as nonlinearities (leaky ReLU), batch normalization, and graph size normalization after each layer. We additionally concatenate all GNN layer outputs as our input to the final linear layer, which empirically helps performance over most tasks. We apply sigmoid activation after the final layer to map the outputs to $[0, 1]$.

As in Wenkel et al. (2025), we use the vertex degrees, local clustering coefficients, and triangle counts as node features. Empirically, this performed superior compared to the Dirac encodings used in Karalias & Loukas (2020) and largely on par with Laplacian and random walk-based encodings. The output of our GNN is a vector $\mathbf{p}$ where $\mathbf{p}_i$ is thought of as the probability that $v_i$ is in the set of interest.[1]

---

[1]For $K$-coloring, we output $K$ vectors where $(\mathbf{p}_k)_i$ is thought

*Table 2.* Performance comparison of GCON with other baselines on **RB-small** datasets (Mean $\pm$ Std). The best deep learning-based method is listed in gold, second best in silver. [†] indicates Gurobi outperforms all deep learning methods.

| METHOD | TYPE | MVC $\downarrow$ | MCLIQUE $\uparrow$ | MIS $\uparrow$ |
|---|---|---|---|---|
| TRUE SIZE | — | 206.95 | 19.07 | 20.07 |
| GUROBI | OR | — | $19.05^\dagger$ | $19.98^\dagger$ |
| GFN | SSL | — | 16.24 | 19.18 |
| GCN | SSL-GNN | $221.56 \pm 0.05$ | $15.33 \pm 0.04$ | $17.67 \pm 0.20$ |
| GIN | SSL-GNN | $221.63 \pm 0.23$ | $15.28 \pm 0.14$ | $17.50 \pm 0.04$ |
| GATV2 | SSL-GNN | $220.76 \pm 2.26$ | $15.56 \pm 0.09$ | $17.58 \pm 0.07$ |
| GCON | SSL-GNN | $211.69 \pm 0.16$ | $16.92 \pm 0.13$ | $18.12 \pm 0.11$ |

### 3.2. Sequential Decoder

The sequential decoder processes the probabilistic output $\mathbf{p}$ of the GNN encoder and enforces the hard problem constraints to arrive at a valid solution. In each case, we order the nodes by probability of being included in the solution set (based on $\mathbf{p}$) and initialize the solution set $S^* = \varnothing$. We then iterate over the ordered node set and add $v_i$ to $S^*$ only if it does not violate the problem constraints.

As demonstrated in Karalias & Loukas (2020), this blueprint follows the method of conditional expectation (Raghavan, 1988). However, we note that this approach does not guarantee the optimal solution, especially when the GNN encoder assigns similar high probabilities to multiple nodes that may constitute distinct solution sets. Therefore, we employ $k$ seeds where each seed selects a different node to initialize the set with (choosing the $k$ nodes with the highest probabilities as initial nodes). We construct $k$ sets, $S_1^*, \ldots S_k^*$, and return the largest or smallest depending on the task. We note that while a similar approach is proposed in Wenkel et al. (2025), their implementation constructed the $k$ sets sequentially, leading to dramatically increased runtimes. By contrast, our sets are constructed in a parallelized manner, enabling the efficient use of higher $k$. We give more details about the decoders in Appendix C.

### 3.3. Objective Functions

As objective functions, we rely on the Ising formulations of the CO problems, as provided in Lucas (2014) and displayed in Table 1. When training our network, we optimize the probabilistic vector $\mathbf{p}$ rather than $X \in \{0, 1\}^N$. The Ising model and equivalent quadratic unconstrained binary optimization (QUBO) formulations have emerged as an effective framework to unify many CO problems by assigning an energy function (namely the Hamiltonian) to every task such that its optimal solution minimizes the total energy (Lucas, 2014; Kochenberger et al., 2014; Glover et al., 2022; Schuetz et al., 2022a). The loss functions for MIS, MDS, MaxClique and MDS involve two terms $A$ and $B$ that can be

---

of as the probability that $v_i$ has color $k$.

tuned additionally; the ratio of $A$ to $B$ controls how strongly to penalize constraint violations in relation to maximizing or minimizing the size of the objective set. Setting $A < B$ ensures the optimal solutions are valid subsets. We note that all six problems studied here (illustrated in Fig. 1) are well established in relevant computer science literature, and their approximate relaxations used to derive the loss terms in Table 1 are becoming increasingly standard in GNN literature. Detailed problem descriptions are provided in Appendix A.

### 3.4. Ablation Study on GCON Layer

To validate our use of GCON in the following studies, we first present an ablation study comparing several GNN-based neural solvers with identical architectures (except the convolutional layer), as well as the performant generative flow network-based GFN model (Zhang et al., 2023) on MVC, MaxClique and MIS. As shown in Table 2, when paired with the energy-based loss functions, we outperform all deep learning baselines (with the exception of GFN on MIS, which is very competitive), attaining particularly strong results for MVC and MaxClique on RB-small graphs. The best-performing GCON models also constitute our pretrained models for our transferability experiments in Sec. 4.

We highlight that, for the MaxClique problem on RB-small, we substantially improve over Wenkel et al. (2025), finding a MaxClique size of 16.92 (compared to 15.87 in Wenkel et al. (2025)) despite the fact that they used a nearly identical architecture. This difference is attributed to our improved loss function that follows the Hamiltonian: By contrast, Wenkel et al. (2025) follows a formulation based on the Motzkin-Strauss theorem (Motzkin & Straus, 1965), which – while also valid and related – uses a quadratic first term that changes the optimization landscape, likely over-emphasizing size maximization over the penalty term. To verify our observations, we further tested a similar quadratic first term for MIS (note the similarity of the MIS and Max-Clique Hamiltonian losses in Table 1); the alternative loss only attained 15.5 average MIS size as opposed to 18.04 with the Hamiltonian loss. These findings strongly emphasize the importance of an appropriate loss function selection, despite the availability of many valid formulations.

## 4. Pairwise Transferability

We begin this section with a brief discussion of known polynomial reductions among three closely related CO tasks: Maximum Independent Set (MIS), Minimum Vertex Cover (MVC) and Maximum Clique (MaxClique).

- **MIS ↔ MVC:** MIS and MVC are trivially reducible as they are the complements of each other by Lemma B.1: Solving for the MIS subset $\text{MIS}_G \subset V$ implies $\text{MVC}_G = V \backslash \text{MIS}_G$, and vice versa.

- **MaxClique ↔ MIS:** Again by Lemma B.1, the Max-Clique of $G$ is the MIS of the complement graph $\text{MaxClique}_G = \text{MIS}_{\bar{G}}$.

- **MaxClique ↔ MVC:** Following the statements above, the MaxClique of $G$ is the complement set of the MVC of the complement graph $\bar{G}$: $\text{MaxClique}_G = V \backslash \text{MVC}_{\bar{G}}$.

### 4.1. MIS ↔ MVC

As established above, $\text{MIS}_G$ and $\text{MVC}_G$ are complements, and therefore solve each other implicitly. In our training pipeline, we can interpret this in terms of the node probabilities, where for a given node $v_i$, we have $\mathbf{p}_i^{\text{MIS}} = 1 - \mathbf{p}_i^{\text{MVC}}$.

We conjecture that the graph representations learned on MIS should also be sufficient to solve MVC, and vice versa. Furthermore, given the linear relationship between the two, a simple linear layer after message-passing should be able to learn this function using identical representations. Our first test is thus to pretrain a model on one task, and transfer the representations to the other task using the pretrained GNN backbone and resetting & training only the linear post-message-passing layer; we expect the transferred model to quickly recover the original pretrained model performance.

Our hypothesis holds true with some caveats, with the results denoted in Table 3. We test several settings, the most straightforward of which is freezing the MIS-trained GNN backbone and simply resetting & fine-tuning the linear output layer to solve MVC, and vice versa. The linear model converges quickly (within 5 epochs for MIS, $\sim 50$ epochs for MVC), but does not match the baselines trained from scratch. This is likely due to the approximate nature of our solution, as the strict duality between the problems assumes an exact solution. In the second setting, instead of initializing the linear output layer from scratch, we invert it, i.e., we initialize the new linear layer by multiplying all parameters by $-1$, as the inversion is maintained through the softmax output: $\sigma(-x) = 1 - \sigma(x)$. This provides a much better initialization as convergence is almost immediate on both tasks, but it does not lead to any substantial improvement on final results – implying that the frozen GNN representations themselves are not able to overcome the duality gap, and thus the backbone also needs to be fine-tuned.

Finally, we verify whether fine-tuning the whole model after inverting the output head can recover the baseline performance. For MIS → MVC in particular, the results are remarkable in that all runs converge within 15 epochs, and the resulting models outperform the baseline trained from scratch for 300 epochs. MVC → MIS is also successful, albeit to a lesser extent – likely because pre-trained MIS model has converged to a better minimum: The fine-tuned model converges to a marginally worse performance than the baseline on average, and convergence takes 100-150 epochs, whereas the baselines were trained for 200 epochs. This

indicates that the GNNs *can* learn to perform the appropriate reduction, but rely on good initialization and fine-tuning of the backbone to maximize performance – theoretical equivariance of the reduction alone is not enough.

## 4.2. MIS/MVC $\leftrightarrow$ MaxClique

Our experiments in 4.1 (MIS $\leftrightarrow$ MVC) implicitly use the fact that the problem reduction does not alter the graph topology. This increases transferability substantially as the graph representations do not suffer from any distribution shift in the graph structure nor the node features, which makes it viable to freeze the GNN backbone, and fine-tuning only the MLP head, in order to obtain close-to-optimal results. Indeed, the learned node-level representations in this case are sufficient, and no additional graph-level information is needed to solve either task.

However, this advantage does not apply to transferring representations between MaxClique and MIS/MVC: $G$ and the complement graph $\bar{G}$ have drastically different structural properties and distributions despite sharing the same node set $V$. For instance, RB graphs are typically sparse, which implies that their complements are very dense. This renders the structurally-derived node features (or any structurally-informed PSE that may be used in their stead for node identifiability) meaningless. Moreover, the graph convolutional layers will similarly suffer from this distribution shift.

With this in mind, we aim to answer the following questions on MIS $\rightarrow$ MaxClique transferability, while noting that our findings apply to MVC $\leftrightarrow$ MaxClique as well:

- **Exps. #4-7:** Despite the shift in graph topology, how useful are the MIS-pre-trained weights by themselves in a (i) frozen backbone, or (ii) full fine-tuning setting?
- **Exps. #8-9:** Assuming the node-level representations are not sufficient, does additional *global* message-passing (using a stack of Graph Transformer (GT) layers on top of the GNN backbone) help adapt to the distribution shift in topology?
- **Exps. #10-11:** Recalling MaxClique$_G$ = MIS$_{\bar{G}}$, can we match baseline performance by implementing the *true* reduction by fine-tuning the model to solve MIS on the

*Table 3.* MIS $\leftrightarrow$ MVC transferability on **RB-small**. BASELINE refers to models trained from scratch on the CO task tested on. Other models are pretrained on the opposite task, e.g., the MIS column evaluates models pretrained on MVC and fine-tuned on MIS.

| GNN | OUT-HEAD | MIS $\uparrow$ | MVC $\downarrow$ |
|---|---|---|---|
| BASELINE | | $18.12 \pm 0.11$ | $211.69 \pm 0.16$ |
| FREEZE | RESET + FT | $17.68 \pm 0.04$ | $212.46 \pm 0.25$ |
| FREEZE | INVERT + FT | $17.69 \pm 0.05$ | $212.39 \pm 0.21$ |
| FT | INVERT + FT | $18.00 \pm 0.05$ | $211.56 \pm 0.24$ |

*Table 4.* Overview of MIS $\rightarrow$ MaxClique transferability experiments on **RB-small**. First two rows represent baselines trained from scratch. The "feats" column denotes whether graph statistics from only the original graph $G$ or also its complement $\bar{G}$ are provided as initial node features. Third row represents a randomly initialized GCON model. Mean $\pm$ standard dev for 3 runs reported. Top 3 transfer runs are denoted with **gold**, **silver** and **bronze**.

| # | FEATS | GNN STACK | GT | COMP? | MC SIZE |
|---|---|---|---|---|---|
| 1 | $G$ | BASELINE | — | FALSE | $16.92 \pm 0.13$ |
| 2 | $G\|\bar{G}$ | BASELINE | — | FALSE | $16.63 \pm 0.05$ |
| 3 | $G$ | RANDOM | — | FALSE | $10.71 \pm 0.14$ |
| 4 | $G$ | FROZEN | — | FALSE | $16.12 \pm 0.20$ |
| 5 | $G$ | FINE-TUNED | — | FALSE | $\mathbf{16.55 \pm 0.03}$ |
| 6 | $G\|\bar{G}$ | FROZEN | — | FALSE | $15.98 \pm 0.12$ |
| 7 | $G\|\bar{G}$ | FINE-TUNED | — | FALSE | $\mathbf{16.63 \pm 0.03}$ |
| 8 | $G$ | FROZEN | 3-MHA | FALSE | $16.13 \pm 0.17$ |
| 9 | $G\|\bar{G}$ | FROZEN | 3-MHA | FALSE | $16.11 \pm 0.63$ |
| 10 | $G\|\bar{G}$ | FROZEN | — | TRUE | $15.52 \pm 0.08$ |
| 11 | $G\|\bar{G}$ | FINE-TUNED | — | TRUE | $\mathbf{16.82 \pm 0.04}$ |

complement $\bar{G}$? To account for the distribution shift in node features, we pre-train and fine-tune the model based on node features derived from *both* $G$ and $\bar{G}$.

## RESULTS & ANALYSIS

In Table 4, we compare our fine-tuned models with the baselines that comprise the first three rows: (1) GCON trained from scratch on MaxClique, using as node features graph statistics derived from $G$ only and representing a typical training setting, (2) GCON similarly trained from scratch but node features derived from both $G$ and $\bar{G}$ for fair comparison with fine-tuning settings that use $\bar{G}$, and (3) Randomly initialized GCON to benchmark random embedding performance. The baselines are trained for 700 epochs, while the frozen/fine-tuned models are trained for 200.

All fine-tuning settings perform well above random, which is an encouraging if unsurprising result. We also note that even the worst-performing models are able to outperform local-message-passing GNNs trained from scratch (Table 2) and find an average maximum clique of size 16 (with the exception of #10). These results convincingly demonstrate that pretraining on MIS is *useful* for MaxClique even though the underlying topology of the MaxClique-reduced-to-MIS graphs are different. We also note that the flexibility of GCON layers likely contributes to the transferability of representations. Nevertheless, most fine-tuning settings fall short of the baselines. We hereby list our main takeaways on each question previously stated:

**Freezing vs. Fine-tuning (#4-7):** The aforementioned distribution shift problems limit the ceiling of the frozen-GNN-backbone models in the absence of additional learnable GNN layers, verifying that the MIS-trained local representations are not sufficient to predict MaxClique. Fine-tuning fares considerably better – while not always sufficient to match the baseline performance with only 200 epochs, #7 ex-

actly matches its corresponding baseline (#2). While more topology-preserving reductions imply easier task transfer (as in the MIS $\leftrightarrow$ MVC case), these experiments show that CO tasks related via non-topology-preserving reductions *can* still provide a strong initialization, with the caveat that fine-tuning is required to close the distribution shift.

**Global message-passing (#8-9):** When considering additional global message-passing (i.e., graph Transformer) we focus on the frozen-backbone setting, since the fine-tuning setting renders them somewhat redundant, especially considering the additional quadratic computational complexity of the GT layers. We use 3 layers of multi-head attention with 3 heads. We also evaluated combining MHA with message-passing using GraphGPS layers (Rampášek et al., 2022), but did not see any additional benefits.

On average, global message-passing over frozen GNN backbones only offers marginal improvements, and falls short of trained baselines. However, we note high variance across runs, with one #9 run attaining 16.75, which outperforms the corresponding baseline (#2). This indicates that global message-passing *may* leverage the frozen MIS representations successfully to transfer to MaxClique, but naïve training of the transformer layers is likely insufficient – extensive fine-tuning and auxiliary structures (e.g., re-introducing node statistics or positional encodings, at the transformer-stack level, or deeper stacks with linear transformer layers) are required to obtain consistent improvements.

**Implementing the true reduction (#10-11):** Finally, we test the following setting: After pretraining our model to solve MIS, in the fine-tuning phase we perform message-passing over the complement graph $\bar{G}$ instead of the original. One can then use the MIS loss and metrics, effectively solving MIS over $\bar{G}$ which is equivalent to solving MaxClique (per the preamble of Sec. 4); note that keeping the Max-Clique loss and metrics over the original graph is also viable as the losses are equivalent over the complement (Table 1).

Interestingly, if the MIS-pretrained backbone is frozen (#12), we obtain by far the worst-performing model, whereas fine-tuning the full model instead (#13) gives us the only model that can consistently match (and even beat) its baseline counterpart within the allotted epochs. The failure of the frozen model can likely be attributed to the distribution shift in the graph topology; however, the learned representations clearly form a useful initialization – fine-tuning quickly adapts the model to the new distribution to recover the baseline performance in less than a third of the epochs.

## 5. Multi-Task Transferability

We now study transferability between graph CO problems in a pretraining-fine-tuning framework, where we learn multiple CO problems simultaneously in pretraining and use this knowledge to transfer efficiently onto new problems by fine-tuning for a few epochs. We first show that a task only requires one task similar to itself in the pretraining set to learn a solution quickly when fine-tuned. We further show that almost all tasks benefit from fine-tuning on other tasks compared to training from scratch, and use our observations to select a pretraining set of 3 tasks and a fine-tuning set of 3 tasks. We then evaluate the resulting performances against single task learning. We hope that a careful study of the interplay between different CO problems and known reductions will yield an appropriate pretraining set, paving the way to foundational models for graph CO problems. For every experiment in this section, we pretrain for 200 epochs on a Barabási–Albert (BA)-small graph dataset and fine-tune on the same dataset for only 20 epochs. The model consists of a GCON backbone where we attach a simple MLP heads for every tasks, and the output is fed to a task specific unsupervised loss function (see Sec. 3.3 and Appendix A). Fine-tuning is done by attaching a new task head with the corresponding loss function and training for 20 epochs with either a frozen or an unfrozen backbone. The tasks considered here are MaxCut, MaxClique, MDS, MIS, MVC and $K$-coloring with $K = 10$. Tables display the size of the subset $V'$ learned for all tasks but $K$-coloring, where we simply count the number of violations.

### 5.1. Leave-One-Out Transfer

We begin by asking ourselves if the successful transfers observed in Sec. 4 also apply in the multi-task learning setting. Suppose we want to fine-tune task $T_n$ on a backbone pretrained on tasks $T_1, \cdots, T_{n-1}$. To get efficient transfer, does it suffice for there to exist one task $T_i$ in the pretraining set such that $T_n$ is reducible to $T_i$? To study this question, we look at MIS and MVC when they are fine-tuned on the other task (MIS from MVC and MVC from MIS), when they are fine-tuned on the rest of the tasks (MaxCut, MaxClique, 10-Coloring and MDS), and when they are fine-tuned on all tasks (rest + other). We compare the results to training MIS and MVC individually for 20 epochs, and show the results for a frozen and an unfrozen backbone in Figure 2.

Unsurprisingly, we find that when we transfer from MVC to MIS and vice-versa, we quickly converge to a good solution (within 4-6 epochs), whereas training from scratch takes at least 15 epochs to match the fine-tuning performance. However, if we do not freeze the backbone, we find that the behavior of the MIS learning curve when pretrained on all other tasks including MVC (and vice versa) behaves similarly to when only pretrained on MVC (MIS resp.), whereas when we remove MVC (resp. MIS) from the pretraining set, the curves behave closer to the training from scratch. Moreover, when the backbone is frozen, the model is not able to transfer to MIS (resp. MVC) using the other tasks if MVC (resp. MIS) is not present in the pretraining set, giving poor

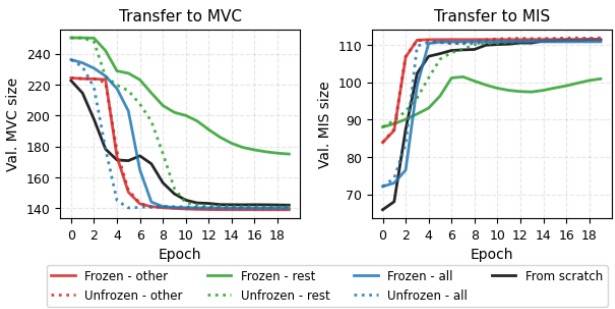

*Figure 2.* Transfer performance on MVC and MIS. We compare training from scratch against frozen and unfrozen fine-tuning on the **other** task (MVC on MIS and MIS on MVC), the **rest** of tasks (MCut, MClique, MDS, $K$-coloring), and **all** tasks (rest + other) for **BA-small** graphs.

*Table 5.* Leave-one-out fine-tuning in low-resource regime on **BA-small** graphs (20 epochs, unfrozen backbone, 5 runs average)

| TASK | FROM SCRATCH | FINE-TUNED |
|---|---|---|
| ↑ MAXCUT | $716.81 \pm 3.00$ | $\mathbf{722.40 \pm 1.17}$ |
| ↑ MAXCLIQUE | $4.31 \pm 0.01$ | $\mathbf{4.32 \pm 0.01}$ |
| ↓ MDS | $\mathbf{35.57 \pm 2.41}$ | $36.15 \pm 1.87$ |
| ↑ MIS | $111.33 \pm 0.20$ | $\mathbf{111.56 \pm 0.10}$ |
| ↓ MVC | $141.30 \pm 0.27$ | $\mathbf{140.04 \pm 0.33}$ |
| ↓ COLOR | $61.92 \pm 36.11$ | $\mathbf{24.19 \pm 9.08}$ |

results, while if MVC (resp. MIS) is present, it actually converges faster than training from scratch. These results suggest that it is sufficient to have one $j \in \{1, \cdots, n-1\}$ such that there is an efficient reduction from $T_n$ to $T_j$ in order to observe rapid transfer from the pretraining set to $T_n$ that beats the baseline in a low-resource regime, even with a frozen backbone (thus requiring to train only a fraction of the total number of parameters).

Now we want set up a leave-one-out transfer framework to study if fine-tuning on a multi-task learning model pretrained on the 5 other tasks help the remaining task to learn a good solution faster than from scratch. We repeat this experiment 5 times for all tasks on BA small graphs in low-resources regime (20 epochs), and display the results in Table 5. We find that for every task but MDS, the model attains better performance at 20 epochs when it is fine-tuned on the other tasks than from scratch, which provides a strong evidence of transfer between graph CO problems. However, the benefits of fine-tuning varies between tasks: it is most beneficial for MaxCut and $K$-coloring to be fine-tuned on other CO problems, whereas MaxClique and MDS show little to no benefit. Therefore if our goal is to rapidly obtain good solutions to CO problems by fine-tuning tasks on a pretrained backbone compared to from scratch, our results suggest that MaxCut and $K$-coloring are good candidates to be fine-tuned.

*Table 6.* MDS (↓) trained together with all but one task on **BA-small** graphs for 200 epochs (5 runs average)

| MAXCUT | MAXCLIQUE | MIS | MVC | COLOR |
|---|---|---|---|---|
| **33.36** | 41.34 | 41.40 | **34.68** | 45.29 |

*Table 7.* Fine-tuning on MDS-MIS-Coloring pretrained backbones for 20 epochs (**BA-small** dataset, 3 runs average)

| PROBLEM | FINE-TUNED | BASELINE | FULL |
|---|---|---|---|
| ↑ MAXCUT | $722.77 \pm 1.00$ | $716.90 \pm 1.55$ | $726.58 \pm 0.51$ |
| ↑ MAXCLIQUE | $4.32 \pm 0.01$ | $4.31 \pm 0.00$ | $4.43 \pm 0.02$ |
| ↓ MDS | $29.65 \pm 0.09$ | $33.93 \pm 1.15$ | $29.56 \pm 0.22$ |
| ↑ MIS | $111.94 \pm 0.06$ | $111.33 \pm 0.26$ | $112.23 \pm 0.06$ |
| ↓ MVC | $139.70 \pm 0.31$ | $141.43 \pm 0.23$ | $139.40 \pm 0.20$ |
| ↓ COLORING | $17.29 \pm 1.97$ | $49.04 \pm 21.84$ | $3.52 \pm 1.52$ |

### 5.2. Fine-Tuning on a Graph CO Backbone

In this section, we propose a pretraining set of three CO problems and leave the remaining three tasks to be fine-tuned for 20 epochs and compare performance to single task learning from scratch for 20 epochs and for the full 200 epochs. From our findings in in Sec. 5.1, we assert that having one closely related task in the pretraining set is sufficient for good fine-tuning. Since we want to manage our resources efficiently, we want to avoid having similar tasks in the pretraining set when only one is sufficient to learn the others by fine-tuning. Therefore, results from Sec. 4 suggest including only one of MIS, MVC and MaxClique in the pretraining set (preferably MIS or MVC), leaving one spot for MDS, $K$-coloring or MaxCut in the fine-tuning only set. Since MDS does not show any gains when fine-tuned on other tasks, unlike MaxCut and $K$-coloring, we choose to include it in the pretraining set. Moreover, we observe that when pretraining the all-but-one multi-task models, MDS performance is best when MaxCut and MVC are not pretrained alongside it, as we show in Table 6, suggesting to leave MaxCut and MVC out of the pretraining set. This implies by our previous remark that we will have MIS in pretraining and MaxClique in fine-tuning-only. We then delegate $K$-coloring to the pretraining set, which is also sensible since it is the most different from the other tasks, being the only one not linearly reducible to 0-1 programming as per Filar et al. (2019) (also see Sec. 2 and Appendix B). In fact, we want to maximize task diversity in the pretraining set, so that the network has access to a wide range of information when fine-tuning. Therefore, we select MDS, MIS, $K$-coloring for pretraining and leave MaxClique, MaxCut, MVC to be fine-tuned. In Table 7 we show the results of fine-tuning all tasks on our backbone, which is unfrozen (except for fine-tuning $K$-coloring), and baseline results from training single tasks from scratch for 20 epochs (BASELINE) and 200 epochs (FULL).

We see that leveraging known polynomial reductions and interactions between CO tasks allows to carefully select a backbone of three tasks, to which we can attach simple task-specific MLP heads and fine-tune to get results that are on par with the fully trained (200 epochs) single task models for most tasks and that beat the low compute resources (20 epochs) singe task models for all tasks. These results suggest that using reductions and interactions between tasks as a guide allows to select an efficient and transferable backbone, which becomes crucial as we increase the number of CO tasks. Therefore, we believe our methodology establishes an important step to building a foundational model for combinatorial optimization problems on graphs.

## 6. Conclusion & Future Work

In this work, we have established a conceptual relation between computational reducibility – a topic traditionally associated with theoretical computer science, and learned transferability – an increasingly prominent area of machine learning in general, and deep learning in particular. We demonstrate that established knowledge of reducibility between problems can help identify and inform pretraining and transfer learning targets, although this relationship is not trivial, and further work remains to be done to fully understand the potential interaction between these notions. After initially exploring pairwise task transferability, we turned our attention to multi-task learning, which would be crucial for the formation of universal models (e.g., implemented via foundation models). We show that, indeed, this is a promising new direction for establishing theory-informed foundation models for graph CO, in which one may find a sufficient suite of landmark tasks that are sufficient for allowing efficient (if not ready-made) transfer of pretrained models to a wide family of significant tasks of interest.

## Acknowledgements and Disclosure of Funding

This work was was partially funded by Bourse en intelligence artificielle des Études supérieures et postdoctorales (ESP) 2023-2024 [Semih Cantürk]; the Fin-ML CREATE graduate studies scholarship for PhD, the J.A. DeSève scholarship for PhD and Guy Wolf's research funds [Frederik Wenkel]; NSF OIA 2242769 [Michael Perlmutter]; Canada CIFAR AI Chair, IVADO (Institut de valorisation des données) grant PRF-2019-3583139727, FRQNT (Fonds de recherche du Québec - Nature et technologies) grant 299376 and NSERC Discovery grant 03267 [Guy Wolf]; NSF DMS grant 2327211 [Michael Perlmutter and Guy Wolf]. This research was also enabled in part by compute resources provided by Mila (mila.quebec). The content provided here is solely the responsibility of the authors and does not necessarily represent the official views of the funding agencies.

## Impact Statement

This paper presents work whose goal is to advance the field of Machine Learning. There are many potential societal consequences of our work, none which we feel must be specifically highlighted here.

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

# A. Graph Combinatorial Optimization Problem Descriptions

We consider a graph $G = (V, E)$ with $|V| = N$. For problems defined on vertex subsets, we let $X \in \{0, 1\}^N$ denote a hard assignment vector where $X_i = 1$ indicates that $v_i$ is in the set of interest. When training our network, we will approximate $X_i$ by a soft assignment vector $\mathbf{p} \in [0, 1]^N$, where $\mathbf{p}_i$ is interpreted as the probability that $X_i = 1$. However, when defining the problems, it is more natural to use the hard assignment vector $X$.

**MaxCut** aims to find a partition $V' \sqcup V'' = V$ maximizing the number of cut edges. We minimize

$$-\sum_{(i,j)\in E} \frac{1 - \sigma_i \sigma_j}{2}$$

where $\sigma_i = 2X_i - 1$. Note that each term vanishes when $v_i$ and $v_j$ are in the same subset and decreases the loss when they are separated.

**Maximum Clique** aims to find the largest subset $V' \subseteq V$ such that all pairs of nodes in $V'$ are connected. We minimize

$$-A\sum_{i=1}^{N} X_i + B \sum_{(i,j)\notin E} X_i X_j$$

where the first term favors large cliques, while the second penalizes missing edges inside $V'$.

**Minimum Dominating Set** aims to find the smallest subset $V' \subseteq V$ such that every node in $V \setminus V'$ has a neighbor in $V'$. We minimize

$$A\sum_{i=1}^{N} X_i + B \sum_{i=1}^{N}(1 - X_i) \prod_{v_j \in \mathcal{N}(v_i)} (1 - X_j)$$

where the first term favors small dominating sets and the second term penalizes nodes that are neither selected nor adjacent to any selected node (with $\mathcal{N}(v_i)$ denoting the immediate neighbors of $v_i$).

**Maximum Independent Set** aims to find the largest subset $V' \subseteq V$ such that no two nodes in $V'$ are adjacent. We minimize

$$-A\sum_{i=1}^{N} X_i + B \sum_{(i,j)\in E} X_i X_j$$

where the first term maximizes the size of $V'$, while the second penalizes edges between nodes of $V'$.

**Minimum Vertex Cover** aims to find the smallest subset $V' \subseteq V$ such that every edge has at least one endpoint in $V'$. We minimize

$$A\sum_{i=1}^{N} X_i + B \sum_{(i,j)\in E} (1 - X_i)(1 - X_j)$$

where the first term minimizes the size of $V'$, and the second term penalizes uncovered edges.

Given $K$ colors, **K-Coloring** aims to assign one color to each node so that all adjacent nodes have different colors. Here, we use an assignment matrix $X \in [0, 1]^{N \times K}$, where $X_{i,k} \approx 1$ indicates that node $i$ has color $k$. We minimize

$$\sum_{i=1}^{N} \left(1 - \sum_{k=1}^{K} X_{i,k}\right)^2 + \sum_{(i,j)\in E} \sum_{k=1}^{K} X_{i,k} X_{j,k}$$

where the first term enforces one color per node, and the second penalizes monochromatic edges. We measure success of this tasks by the number for violations observed, i.e. the number of edges connecting two vertices of the same color is to be minimized.

# B. Extended Related Work

In this section we provide an extended discussion of the related work that was partially introduced in Sec. 2.

The discrete nature of combinatorial optimization problems and their natural representability through graphs render them of particular interest to the graph learning community. As part of the recent interest in neural CO solvers, several prominent lines of research emerged. Most early prominent works, such as Gasse et al. (2019); Li et al. (2018) and Selsam et al. (2019), have focused on *supervised* approaches for neural CO. However, supervised approaches are inherently limited in that obtaining labels for NP-hard problems becomes computationally infeasible. Consequently, other lines of works have taken precedence in recent years, namely reinforcement learning (RL) (Dai et al., 2017; Bello et al., 2017; Yolcu & Poczos, 2019) (and related methods like generative flow networks (Zhang et al., 2023)), and unsupervised learning (Tönshoff et al., 2021; 2023; Amizadeh et al., 2019; 2020; Sanokowski et al., 2024). Unsupervised methods are typically concerned with designing task-specific loss functions such that the optimal solution to the CO problem minimizes this loss.

Our work is closely related to a family of methods that train graph neural networks (GNN) in an unsupervised, end-to-end differentiable manner. Karalias & Loukas (2020) spearheaded this direction, training a GNN to learn a distribution over possible node subsets using surrogate loss functions for maximum clique and graph partitioning, and sequentially decoding the probabilistic predictions to obtain valid solutions. Min et al. (2022) pair this framework with a graph convolutional layer derived from the graph scattering transform (Gao et al., 2019) to obtain further competitive results on maximum clique, while Min et al. (2023) apply a similar setup for the traveling salesman problem (TSP).

Finally, a subset of these unsupervised GNN-based solvers is of particular interest to us due to their choice of *physics-inspired* loss functions. Specifically, Lucas (2014) demonstrate how many CO problems map to the Ising model from physics with a corresponding energy landscape represented by a Hamiltonian; minimizing the Hamiltonian is then equivalent to optimizing the CO in the form of a quadratic unconstrained binary optimization (QUBO) problem. Schuetz et al. (2022a); Krylova & Phillipson (2025) tackle MaxCut and MIS using this recipe while Schuetz et al. (2022b) extend it to graph coloring. More recently, Sanokowski et al. (2024) uses the task-specific Hamiltonians to train a diffusion model to attain competitive results across multiple tasks.

**Multi-task Neural CO.** Most attempts to develop a unified model for combinatorial optimization problems are based on a supervised learning (SL) or a reinforcement learning (RL) framework. For example, Drakulic et al. (2024) proposes GOAL, a generalist model consisting of a single backbone based on mixed-attention blocks that learns from imitating expert trajectories, and allows to solve multiple CO problems and fine-tune on new ones. Similarly, Zong et al. (2025) proposes UniCO, a unified model for solving different CO problems by framing each problem as a Markov Decision Process and incorporating a CO-prefix design and a two-stage self-supervised learning scheme. Moreover, Li et al. (2025); Wang et al. (2025) propose RL-based head-encoder-decoder models using gradient homogenization and a multi-armed bandit algorithm respectively to optimize multi-task learning of CO problems. Other methods (Berto et al., 2024; Lin et al., 2024; Liu et al., 2024; Zhou et al., 2024) propose RL-based pretraining-fine-tuning approaches specifically for different variations of vehicle routing problems (VRCs). Finally, some authors try to learn a generic representation for graph CO problems by converting these problems to bipartite SAT graphs (Guo et al., 2025; Zeng et al., 2023) or constructing graphs by breaking down constraints of a problem into an abstract syntax tree and expressing relationships through the edges (Boisvert et al., 2024). Unlike any of these methods, we rely on an unsupervised learning approach that employs task-specific loss functions, which avoids problems arising from the large size of the state space in RL. Moreover, we avoid transforming the graphs to a common SAT or abstract representation but rather work on the original graph structure by using a GCON backbone that leverages a complex filter bank and localized attention mechanisms to learn approximating solutions to CO problems (Wenkel et al., 2025). Finally, unlike most of the above methods which do not justify their selection of pretraining tasks, we aim to use known polynomial reductions to show similar problems transfer effectively between each other and then to guide our pretraining tasks selection.

**Polynomial reductions for CO problems.** Combinatorial optimization problems have been extensively studied in the computer science literature, for example with Karp's 21 NP-complete problems (Karp, 1972). Many polynomial reductions are well known between these problems. For instance, reductions between the maximum clique (MaxClique), maximum independent set (MIS) and minimum vertex cover (MVC) can be obtained from the following lemma.

**Lemma B.1** (Garey & Johnson, 1979). *For any graph $G = (V, E)$ and subset $V' \subseteq V$, the following statements are equivalent: 1. $V'$ is a vertex cover for $G$; 2. $V \setminus V'$ is an independent set for $G$; and 3. $V \setminus V'$ is a clique in the complement $\bar{G}$ of $G$, where $\bar{G} = (V, \bar{E})$.*

Moreover, Filar et al. (2019) demonstrate the existence of a kernel subset of Karp's 21 problems. Specifically, they define problem $Q$ as belonging to the linear orbit of $P$ if there exists a reduction where the input size of $Q$ is bounded by a linear function of the input size of $P$. They then show that every problem in Karp's original set can be reduced with linear growth in problem size to at least one problem within this kernel. In particular, their work implies that Max-Clique, Max-Cut, Node Cover (MVC and therefore MIS), and Set Cover (of which MDS is a special case) reside in the linear orbit of 0-1 Programming. Note that conversely, problems like Chromatic Number ($K$-coloring) and the Hamiltonian Cycle Problem appear to occupy different linear orbits.

## C. Experimental Setting

**Datasets.** We conduct our experiments on the same synthetic datasets used in Wenkel et al. (2025), which are commonly used in the field of neural graph combinatorial optimization. In all experiments, we use datasets of size 6000 with 200-300 nodes (small) and 800-1200 nodes (large). In Section 4, we use graphs generated from the RB model. For RB graphs, we can specify the number of cliques ($n$) and the number of nodes per cliques ($k$). We use $n \in [20, 25]$ and $k \in [5, 12]$ for RB-small and $n \in [40, 55]$ and $k = 20$ for RB-large. We then use Barabási–Albert (BA) graphs in Section 5. For BA graphs, we can specify how many edges are attached from each new node, which we set to 4 for both BA-small and BA-large.

**Backbone architecture.** We use a GCON backbone based on Wenkel et al. (2025) with the specific configuration detailed in Table 8 for every experiments. Total number of parameters depend on the size of the head used, but the GCON backbone always has 39.2K parameters.

*Table 8.* GCON Backbone and Training Hyperparameters

| Parameter | Value |
| --- | --- |
| *Architecture Details* | |
| Number of GCON Layers | 16 |
| Hidden Dimension ($d_h$) | 64 |
| Channels (Filters) | [0], [1], [2], [4], [0,1], [1,2], [2,4] |
| Aggregation | Sum |
| Dropout Rate | 0.3 |
| Backbone Activation | ELU |
| Pre-MP / Post-MP Layers | 1 / 1 |
| Node Encoder Stats | Degree, Cluster Coeff., Triangles |
| *Optimization & Training* | |
| Optimizer | AdamW ($\eta = 10^{-3}$, $w_d = 0$) |
| Learning Rate Scheduler | Cosine with Warmup (50 epochs) |
| Batch Size | 256 (small) or 128 (large) |
| Loss Strategy | Weighted Sum (Task weight $= 1.0$) |

**Decoders.** The candidate sets are constructed by first ranking all vertices in descending order of their model-predicted scores. For each problem type, the algorithm generates K distinct candidate sets (seeds) in parallel to find the best possible solution for that specific graph. In the MaxClique and MIS decoders, the k-th candidate set is initialized with the k-th ranked node, and the algorithm then greedily attempts to add subsequent nodes in the ranked list only if they respect the problem's constraint, such as forming a clique or an independent set. For the MDS and MVC, the k-th candidate set is constructed by skipping the top k-1 ranked nodes (by setting them to ) and then greedily selecting the highest-ranked available nodes until the entire graph is covered. Maxcut simply picks nodes with a score higher than 0.5 to be in the first set and the rest in the second and counts the number of cuts created by this partition, while coloring simply counts the violations (adjacent nodes of the same color).

## D. Multi-Task Learning vs. Unified-Task Reductions

A related but distinct set of approaches in neural CO instead focus on generalized formulations of CO problems in the form of *satisfiability problems* (SAT) or their optimization counterparts *Max*-SAT. This line of work emerges from the Cook-Levin theorem (Cook, 1971; Karp, 1972), which shows that any problem in NP can be reduced to a boolean satisfiability problem (SAT) in polynomial time.

*Table 9.* Average number of nodes and edges for the original BA-small graph where we train on solving MDS, and the equivalent graphs for other tasks obtained by applying reductions to the original graphs. For the transformed graphs, we denote the % change in the "blow-up" in the nodes and edges of the resulting graph.

| REDUCTION | # NODES | × MDS SIZE | # EDGES | × MDS SIZE |
|---|---|---|---|---|
| MDS (BASE) | $250.3 \pm 28.9$ | – | $985.0 \pm 115.8$ | – |
| MIS | $1239.0 \pm 146.8$ | 4.95 | $2964.0 \pm 352.3$ | 3.01 |
| MVC | $1239.0 \pm 146.8$ | 4.95 | $2964.0 \pm 352.3$ | 3.01 |
| MAXCLIQUE | $30873.7 \pm 7148.0$ | 123.37 | $91870.3 \pm 21357.4$ | 93.27 |

*Table 10.* Comparison of our MTL framework with unified-MDS-reduction-based approaches, where we train a GCON model to solve MDS on BA-small graphs, and then "transform" the graphs corresponding to MIS/MVC/MaxClique via constructing the true reductions over the graph. We test three variants of the pre-trained MDS model: 0-shot, frozen backbone (F-BB), and full fine-tuning (FT). * denotes the original MDS pre-training result, which is trained for 100 epochs as opposed to constraining to 20 epochs for other tasks.

| METHOD | MDS $\downarrow$ | MVC $\downarrow$ | MIS $\uparrow$ | MAXCLIQUE $\uparrow$ |
|---|---|---|---|---|
| (MDS RED.) RANDOM | $120.43 \pm 25.3$ | $190.11 \pm 36.9$ | $59.87 \pm 40.1$ | $1.61 \pm 1.04$ |
| (MDS RED.) BASE (20 EPOCHS) | $31.37 \pm 0.62$* | $165.21 \pm 0.96$ | $80.99 \pm 6.42$ | $2.98 \pm 0.12$ |
| (MDS RED.) 0-SHOT | $31.37 \pm 0.62$* | $148.21 \pm 2.53$ | $102.78 \pm 2.53$ | $2.46 \pm 0.44$ |
| (MDS RED.) F-BB | - | $144.59 \pm 1.35$ | $106.25 \pm 1.35$ | $2.27 \pm 0.14$ |
| (MDS RED.) FT | - | $142.79 \pm 1.37$ | $106.45 \pm 0.92$ | $2.82 \pm 0.01$ |
| (MTL) FT (W/ GRAPH STATS) | $32.03 \pm 0.66$ | $\mathbf{139.31 \pm 0.17}$ | $\mathbf{111.96 \pm 0.02}$ | $\mathbf{4.34 \pm 0.01}$ |

Zeng et al. (2023) pre-train GNNs on bipartite graph representations of synthetic Max-SAT instances in a supervised manner and solve MaxCut, MIS and MDS instances by reducing them to Max-SAT graphs. Guo et al. (2025) follow a similar recipe but focus on SAT formulations of corresponding decision problems, and pre-train in an unsupervised manner via contrastive learning. These works acknowledge the value of generalization for neural CO and provide partial solutions, but do not directly address the relationship between reducibility and transferability; instead, they aim to wholly *circumvent* task transferability itself by directly learning in the general SAT/Max-SAT space, to which all our CO problems of interest can be polynomially reduced.

These approaches, however, give rise to different limitations on transferability over *graph* distributions, rather than over tasks. Specifically, the approaches above require applying the true reduction over the graph of interest, similar to the MIS $\leftrightarrow$ MC reduction (Sec. 4.2) where we demonstrate that finding the maximum clique of $G$ is equivalent to finding the MIS of the complement $\bar{G}$. The graph reductions between MIS, MVC and MaxClique, however, lead to relatively *mild* structural changes, as the node set $V$ is maintained. However, as discussed in Sec.4, even such mild changes in structure lead to tangible limitations on transferability in constrained fine-tuning settings.

The reductions involved to map an, e.g., MaxClique graph to an equivalent Max-SAT graph, on the other hand, require "chaining" multiple structure-altering reductions to arrive at the Max-SAT representations, which in turn leads to extreme blow-ups in the number of nodes and edges considered. For example, reducing MaxCut on a graph to its SAT counterpart, in practice, requires chaining three reductions: MaxCut $\rightarrow$ MIS $\rightarrow$ 3-SAT $\rightarrow$ SAT.

This resulting blow-up, despite being polynomially bounded, leads to a large distribution shift in the graph distributions across which the models are trained and transferred to, as well as substantially larger graphs in general, limiting the ability of GNN-based frameworks to learn the appropriate graph representations effectively. To demonstrate the potential shortcomings of these methods compared to our multi-task learning (MTL) framework, we hereby present an analogous study where we compare our MTL framework with a "unified-task" learning framework akin to the works discussed above.

**Unified-task method.** We pre-train a GCON model to solve MDS on BA-small graphs, and evaluate the transferability on the resulting model on the *reduced* versions of the BA-small graphs for MIS, MVC and MaxClique problems, respectively. The reducibility relationships are defined as

$$\text{MDS} \leftarrow \text{MVC} \leftrightarrow \text{MIS} \leftrightarrow \text{MaxClique}$$

where each arrow indicates whether a known reduction exists in the specified direction. We choose MDS as our "base" task as all other tasks can be reduced to MDS, whereas no established direct reduction from MDS to the other tasks exists to our knowledge without reducing it to a more general CO framing (e.g. Max-SAT). In addition to the MVC $\leftrightarrow$ MIS and MIS $\leftrightarrow$ MaxClique reductions we establish in Sec.4, we introduce the following reduction (Sipser, 1996):

---

**Algorithm 1** MDS $\leftarrow$ MVC

---

**Require:** Undirected graph $G := (V, E)$
  $G' := (V', E') \leftarrow G$
  Remove all nodes with no incident edges from $G'$
  **for** $(i, j) \in E$ **do**
    $V' \leftarrow V' \cup \{k\}$ {For every original edge $(i, j)$, add a new "gadget" node $k$ to $V'$}
    $E' \leftarrow E' \cup \{(i, k), (j, k)\}$ {Connect the new node $k$ to both endpoints of the original edge}
  **end for**
  **return** $G'$

---

The MVC on $G$ is thus equivalent to the MDS on $G'$. The chaining of reductions, in the case of MaxClique $\rightarrow$ MDS, is then as follows:

1. Convert the original graph $G_{\text{MaxClique}}$ to its complement to define the equivalent MIS graph $G_{\text{MIS}} = G_{\text{MVC}}$.
2. Apply 1 to obtain the MDS graph $G_{\text{MDS}}$.
3. Solve the MDS problem on $G_{\text{MDS}}$.
4. Invert the solution set to account for the MVC $\leftrightarrow$ MIS reduction (which does not transform the graph but inverts the node assignments). The resulting set (and thus its cardinality) is equivalent to the MaxClique of $G_{\text{MaxClique}}$.

We follow this recipe to generate reduced versions of our BA-small dataset for MIS, MVC and MaxClique respectively. The node and edge counts of each reduced dataset is presented in Table 9: The first graph reduction from MDS to MIS/MVC leads to graphs with five times the number of nodes and three times the number of edges. The chained reduction to MaxClique leads to an average graph size of about 31,000 nodes, a 122-fold increase over the relatively small (200-300-node) original graphs.

We then evaluate our MDS pre-trained model on each reduced dataset in three transfer settings: zero-shot transfer, fine-tuning the linear prediction head while keeping the backbone frozen, and fine-tuning the whole model. In line with our MTL experiments, we restrict fine-tuning to 20 epochs for a fair comparison.

**Results.** We present the results of our study in Table 10. Our MTL method consistently outperforms all variants of the unified-MDS-task model, with the performance gap growing as we chain more reductions and increase the distribution shift. For MVC and MIS, unified-task pretraining is still useful despite falling short of MTL performance, with noticeable gains even in the zero-shot setting compared to the baseline of training 20 epochs from scratch, with additional improvements through fine-tuning.

However, in the MaxClique case with a $\sim 100\times$ blow-up in the node and edge counts, unified-task pre-training breaks down: Even in the full fine-tuning setting, the attained maximum clique is lower than the baseline, indicating negative transfer. The gap between fine-tuned unified-task and MTL methods also increase from $2.5\%$ and $4.9\%$ for MVC and MIS, respectively, to $35.0\%$ for MaxClique.

**Analysis.** These results suggest a direct inverse correlation between increased distribution shift as a result of chained reductions, and reduced transferability. In addition to the distribution shift itself, the induced large graphs also limit the effectiveness of fine-tuning by introducing practical and computational bottlenecks that reduce the quality of learned graph representations. The practical bottlenecks are perhaps straightforward to analyze: Significantly larger (and denser in cases where the MIS $\leftrightarrow$ MaxClique reduction, which returns dense complements of sparse graphs, is considered) graphs render efficient training of GNNs expensive, require significantly more memory and compute, and provide noisier gradients due to being limited to small batch sizes. Computing graph statistics or positional encodings (PE) for graph datasets with sizes in the order of tens of thousands also render their usage as informative node features cumbersome at best, and non-viable at

*Table 11.* Fine-tuning on MDS-MIS-Coloring pretrained backbones for 20 epochs (**BA-large** dataset, 3 runs average)

| PROBLEM | FINE-TUNED | BASELINE | FULL |
|---|---|---|---|
| ↑ MAXCUT | $2935.82 \pm 0.63$ | $2926.39 \pm 2.46$ | 2951.69 |
| ↑ MAXCLIQUE | $4.31 \pm 0.03$ | $4.32 \pm 0.01$ | 4.36 |
| ↓ MDS | $109.72 \pm 0.16$ | $129.64 \pm 12.74$ | 113.19 |
| ↑ MIS | $453.51 \pm 0.17$ | $452.59 \pm 0.07$ | - |
| ↓ MVC | $550.55 \pm 0.36$ | $554.53 \pm 1.22$ | - |
| ↓ COLORING | $31.40 \pm 5.29$ | $53.48 \pm 4.53$ | 7.61 |

*Table 12.* Size Extrapolation Comparison on BA-large and BA-small datasets (3 runs average)

| PROBLEM | BA-LARGE | | | | BA-SMALL | | | |
|---|---|---|---|---|---|---|---|---|
| | FINE-TUNED | EXTRAPOL. | DIFF | BASELINE | FINE-TUNED | EXTRAPOL. | DIFF | BASELINE |
| ↑ MAXCUT | $2935.8 \pm 0.6$ | $2936.0 \pm 0.1$ | +0.01% | $2926.4 \pm 2.5$ | $722.8 \pm 1.0$ | $722.0 \pm 1.4$ | -0.11% | $716.9 \pm 1.5$ |
| ↑ MAXCLIQUE | $4.31 \pm 0.03$ | $4.30 \pm 0.02$ | -0.24% | $4.32 \pm 0.01$ | $4.32 \pm 0.01$ | $4.34 \pm 0.01$ | +0.49% | $4.31 \pm 0.00$ |
| ↓ MDS | $109.7 \pm 0.2$ | $111.9 \pm 0.8$ | +2.01% | $129.6 \pm 12.7$ | $29.6 \pm 0.1$ | $29.2 \pm 0.1$ | -1.61% | $33.9 \pm 1.2$ |
| ↑ MIS | $453.5 \pm 0.2$ | $453.9 \pm 0.2$ | +0.08% | $452.6 \pm 0.1$ | $111.9 \pm 0.1$ | $111.8 \pm 0.0$ | -0.10% | $111.3 \pm 0.3$ |
| ↓ MVC | $550.5 \pm 0.4$ | $550.9 \pm 0.3$ | +0.06% | $554.5 \pm 1.2$ | $139.7 \pm 0.3$ | $139.8 \pm 0.2$ | +0.09% | $141.4 \pm 0.2$ |
| ↓ COLORING | $31.4 \pm 5.3$ | $53.8 \pm 12.5$ | +71.26% | $53.5 \pm 4.5$ | $17.3 \pm 2.0$ | $28.3 \pm 8.8$ | +63.52% | $49.0 \pm 21.8$ |

worst. We emphasize that we have considered BA-small graphs precisely for these reasons: Despite the small size of the original graphs, the resulting MaxClique dataset is comparatively so large that when training on a single NVIDIA L40S GPU, we can fit only four graphs into memory at a time and full fine-tuning (20 epochs) takes about two days, whereas training the MDS model on the original BA-small dataset only takes several hours.

Learning non-local relationships on large graphs are also more prone to run into under-reaching and over-squashing (Alon & Yahav, 2021), well-known challenges in graph representation learning that limit the expressivity of the learned graph representations. Without auxiliary structures like virtual nodes, skip connections or graph Transformer layers, we conjecture that it is difficult for conventional GNNs to fully overcome these bottlenecks both when training from scratch and in fine-tuning, leading to both the baseline and unified-task models falling well short of our MTL model which instead solves the respective tasks on the original graph.

# E. Additional Experiments

**Multi-task learning on large graphs.** We test our pretraining-finetuning framework developed in Section 5 on BA-large graphs. We see that the results obtained and the observations made in Section 5 also hold for larger graphs.

**Size extrapolation.** We investigate if finetuning on graphs of a certain size allows us to quickly finetune to graphs of different sizes never seen in pretraining. Using the same set-up as Section 5, we pretrain on BA-small graphs and finetune on BA-large graphs, and vice-versa. We observe very good size extrapolation, where for most tasks pretraining on small graphs then finetuning on larger graphs and vice-versa performs as good as pretraining and finetuning on graphs of the same size. Only coloring doesn't transfer as well, where we see that extrapolating from large to small graphs gives better results than training from scratch but not nearly as good as pretraining from same size graphs, while extrapolating from small to large graphs gives similar results to training from scratch.

**Timing experiments.** We record the required time for inference at test time once we have pretrained our model as done in Section 5 to finetune on our 6 different CO problems. We use one NVIDIA L40S GPU for each tasks for fair comparison. Pretraining on single tasks take from a few minutes (MaxCut, Coloring) to a few hours (backbone, MDS, MaxClique).

We also provide the inference times for Table 2 over the test set (500 graphs, NVIDIA Tesla V100) are provided below. Note that despite comparable times to the other methods on the RB-small dataset, GFN and Gurobi scale substantially worse compared to the GNN-based methods on larger graphs. MVC times for Gurobi and GFN are not listed, but would be approximately equal to MIS as one would solve for MIS and invert the solution.

*Table 13.* Inference Time Performance on NVIDIA L40S GPU (seconds)

| PROBLEM | BA-LARGE | BA-SMALL |
|---|---|---|
| MAXCUT | 6.23 | 2.23 |
| MAXCLIQUE | 48.95 | 32.58 |
| MDS | 46.42 | 10.65 |
| MIS | 65.11 | 22.78 |
| MVC | 137.52 | 31.39 |
| COLORING | 6.38 | 2.84 |

| Method | Type | MVC Time | MClique Time | MIS Time |
|---|---|---|---|---|
| True Size | — | — | — | — |
| Gurobi | OR | — | 1:55 | 47:34 |
| GFN | SSL | — | 0:42 | 0:32 |
| GCN | SSL-GNN | 1:29 | 0:57 | 0:45 |
| GIN | SSL-GNN | 1:24 | 0:55 | 0:35 |
| GATv2 | SSL-GNN | 1:26 | 1:02 | 0:57 |
| GCON | SSL-GNN | 1:46 | 1:20 | 1:05 |

*Table 14.* Inference times of Table 2 over the test set (500 graphs, NVIDIA Tesla V100).

**Zero-shot performance on DIMACS instances.** We investigate how pretraining using our method allows us to extrapolate to new graph distribution by evaluating on DIMACS instances, which are a standart reference point for many combinatorial optimization problems. As DIMACS instances consist of specific graph instances and not a standard graph dataset adaptable to our framework, we report inference-only, zero-shot transfer results by running our pretrained model on select DIMACS instances (treating them as a test set) to demonstrate that the learned representations produce reasonable solutions on out-of-distribution real-world graphs. Using the evaluation framework inspired from Feng et al. (2026), we do this for the MIS task by finding the MIS of the complement of DIMACS MaxClique graphs, which is equivalent to the MaxClique of the original DIMACS graphs for which the best known sizes are available (2nd DIMACS Challenge, Johnson & Trick (1996)). Comparing our pretrained model to random initialization, we observed that it comfortably outperforms random models on almost all graphs, and obtains reasonable zero-shot results on many instances, showing that we are able to extrapolate useful information from pretraining on RB-graphs to considerably out-of-distribution hard graph instances. Note we are not expected to attain near-SOTA results when zero-shot learning on considerably out-of-distribution hard single graph instances using an ML-based solver, compared to SOTA algorithmic solvers specifically designed for MIS.

| Graph name | Pretrained | Random | Best known |
|---|---|---|---|
| C1000.9 | 56 | 29 | 68 |
| C125.9 | 32 | 25 | 34 |
| C2000.5 | 11 | 9 | 16 |
| C2000.9 | 48 | 28 | 80 |
| C250.9 | 36 | 28 | 44 |
| C400.5 | 10 | 8 | 18 |
| C500.9 | 50 | 30 | 57 |
| DSJC1000.5 | 11 | 8 | 15 |
| DSJC500.5 | 10 | 8 | 13 |
| MANN_a27 | 117 | 85 | 126 |
| MANN_a45 | 276 | 71 | 345 |
| MANN_a81 | 300 | 122 | 1100 |
| brock200_2 | 10 | 8 | 12 |
| brock200_4 | 13 | 11 | 17 |
| brock400_2 | 21 | 16 | 29 |
| brock400_4 | 23 | 15 | 33 |
| brock800_2 | 16 | 12 | 24 |
| brock800_4 | 16 | 12 | 26 |
| gen200_p0.9_44 | 37 | 28 | 44 |
| gen200_p0.9_55 | 38 | 30 | 55 |
| gen400_p0.9_55 | 48 | 30 | 55 |
| gen400_p0.9_65 | 44 | 31 | 65 |
| gen400_p0.9_75 | 45 | 33 | 75 |
| hamming10-4 | 27 | 16 | 40 |
| hamming8-4 | 16 | 16 | 16 |
| keller4 | 9 | 8 | 11 |
| keller5 | 18 | 15 | 27 |
| keller6 | 37 | 28 | 59 |
| p_hat1500-1 | 10 | 3 | 12 |
| p_hat1500-2 | 52 | 4 | 65 |
| p_hat1500-3 | 79 | 10 | 94 |
| p_hat300-1 | 7 | 5 | 8 |
| p_hat300-2 | 24 | 10 | 25 |
| p_hat300-3 | 31 | 17 | 36 |
| p_hat700-1 | 8 | 3 | 11 |
| p_hat700-2 | 38 | 5 | 44 |
| p_hat700-3 | 57 | 10 | 62 |

*Table 15.* Pretrained (RB small) and random model inference on DIMACS instances for MIS

