# OpenReview forum: "Can Computational Reducibility Lead to Transferable Models for Graph Combinatorial Optimization?"
_ICML.cc/2026/Conference — ICML 2026 regular_

### Official Review · Reviewer_dqgc · 2026-03-12

**Soundness:** 1
**Presentation:** 2
**Significance:** 2
**Originality:** 2
**Overall Recommendation:** 3
**Confidence:** 5

**Summary:**

The paper provides an empirical study evaluating how existing neural frameworks perform when transferred between CO tasks that are closely related and known to be reducible. The authors use GCON as the GNN backbone with an energy-based unsupervised loss function and conduct experiments on transfer learning, including pairwise task transferability and multi-task learning, among a limited set of closely related CO tasks (e.g., MIS, MVC, and Max Clique) on small-scale datasets.

**Compliance With Llm Reviewing Policy:**

Affirmed.

**Final Justification:**

The rebuttal has partially resolved my concerns. I have increased my score for this paper. However, I would like to maintain my negative recommendation for this paper since it still has unresolved issues with empirical evaluation and novelty.

**Key Questions For Authors:**

Please refer to weaknesses W1-W7.

**Limitations:**

No. Overall, the paper does not introduce new techniques and is essentially an empirical study of transfer learning on CO. The authors only conduct experiments on a few closely related CO tasks that are known to be reducible, and the experiments are conducted only based on a single existing neural framework on small-scale datasets. The authors should conduct experiments on a broader range of CO problems and datasets, and provide a more comprehensive evaluation of the transferability of different existing CO frameworks. Moreover, the authors should provide theoretical analysis and significantly improve the writing.

**Strengths And Weaknesses:**

Strengths
S1. The paper clearly explains the motivation behind the framework.
S2. The experimental results are analyzed in detail.

Weaknesses:
W1. The applicability of the proposed transfer learning framework is limited. It appears that the proposed framework can only transfer between a few similar CO problems that are closely related and and already known to be reducible according to prior works (e.g., between MIS and MVC), rather than across diverse types of combinatorial optimization problems (e.g., between TSP and MIS). This significantly restricts the practical impact of the framework. For example, MVC and MIS are already well-known to be complementary problems (i.e., MVC=V/MIS), and therefore it is not surprising to see the transfer learning between these two tasks. The authors should conduct experiments on transfer learning between more diverse types of CO problems (e.g., routing problems and graph similarity computation).

W2. The paper is closer to an empirical study and does not introduce any novel techniques. The proposed framework mainly integrates several existing techniques from prior works in this field:
    W2.1. The GNN encoder (GCON) in Section 3.1 is adopted from [1].


    W2.2. The sequential decoder in Section 3.2 follows a standard sampling-based decoding strategy that has been widely used in many combinatorial optimization frameworks (e.g., [2][3][4]).

    W2.3. The objective function in Section 3.3 is directly adopted from [5].

    W2.4. The polynomial reductions between CO tasks are also adopted from several prior works.

    W2.5. The pre-training and fine-tuning pipeline follows standard training strategies, and the paper does not introduce any additional technical improvements beyond these common practices.

    As a result, the contribution of the paper only lies in the empirical evaluation of transfer on only a small number of closely related CO tasks, rather than in introducing new algorithmic or modeling techniques.

W3. The paper studies the transferability of neural models across closely related CO tasks that are reducible, which should be closely related to theoretical aspects. However, it only provides empirical analysis without any theoretical analysis.

W4. The writing and the presentation of this paper is unclear:
    W4.1. The experimental section is not clearly separated.

    W4.2. The paper organization is somewhat confusing. In particular, it is unusual that the ablation study on the use of GCON is placed in the Methodology section (Section 3). Moreover, it is located in Section 3.4, which is separated from the description of GCON in Section 3.1. At the very least, this ablation study should be placed together with the description of GCON in Section 3.1, rather than appearing later as a separate subsection in Methodology.

    W4.3. Please provide a subsection that clearly describes the implementation details and the test environment.

    W4.4. Please provide a subsection that clearly describes the datasets (e.g., instance sizes, number of instances).

W5. The paper does not introduce new techniques and mainly studies the transferability of existing neural frameworks between closely related tasks. However, the experiments are conducted based on only a single neural framework, and no baseline methods are included for a clear reference. The authors should consider including more types of neural frameworks (e.g., generative CO models, other network architectures designed for CO, different decoding strategies, and different training objectives) and evaluating the transferability of each framework to provide a more comprehensive study.

W6. Transfer learning is typically useful for large-scale instances, where training models from scratch can be computationally expensive on large-scale instances. However, the paper does not provide any evaluation on large-scale graphs and only reports results on the RB-Small and BA-small dataset.

W7. Please also provide the training time under different settings.

[1] Towards a general recipe for combinatorial optimization with multi-filter gnns
[2] DIMES: A Differentiable Meta Solver for Combinatorial Optimization Problems
[3] DIFUSCO: Graph-based Diffusion Solvers for Combinatorial Optimization
[4] Generalize a Small Pre-trained Model to Arbitrarily Large TSP Instances
[5] Ising formulations of many np problems.

---

> ### Author Rebuttal · Authors · 2026-03-31
>
> W1) The objective of Section 4 is to leverage well-known simple reductions to study the relationship between reducibility and transferability. As noted in the paper, good transfer is expected: we conjecture that graph representations learned for MIS are sufficient for MVC and vice versa. Given their linear relationship, a simple linear layer post-message-passing should recover original performance quickly. However, we demonstrate that while simple reductions facilitate efficient transfer, the process is non-trivial (see the discussion on the duality gap with Reviewer BXjf W1 for example). Specifically, fine tuning with a frozen backbone often leaves performance on the table because we transfer from an approximation; since strict duality assumes exact solutions, fine tuning the backbone is necessary to bridge this gap or adapt to distribution shifts (e.g., between MIS and MaxClique). Section 5 then extends this study to a diverse enough set of 6 CO problems, as we argue in response to Reviewer iW2r (W2).
>
> W2) We do not claim a groundbreaking novel architecture, but rather an integration of different Graph Neural CO approaches into a novel MTL framework for CO on graphs. The focus remains on the link between reducibility and transferability. To our knowledge, this is the only MTL/transfer learning method operating directly on graph structures within a purely unsupervised framework, which gives novelty to our framework. We achieve this by adapting a SOTA GNN (GCON) to the MTL setting and improving it via a superior loss function based on unified Ising/QUBO formulations and parallelized decoders.
>
> W3) We provide a theoretical discussion of our experimental insights. Although the problems in Section 4 are theoretically reducible, our results show that transfer is not always seamless, as mentioned in response to W1. This work elucidates that current notions of reducibility do not perfectly align with empirical transferability. While we recognize this is a nascent direction requiring further empirical groundwork before concrete theorems can be formulated, our discussion provides the necessary theoretical scaffolding for these insights.
>
> W4) 1 & 2): We placed the ablation study before the core experiments to justify selecting GCON over popular alternatives like GFN or other local message-passing GNNs. This section establishes a performance baseline for the transfer learning analysis in Section 4. We are happy to move this to Section 3.1 in the final version for better flow. 3): We utilize a GCON backbone based on [Wenkel et al].. The backbone (mp) consistently comprises 39.2K parameters, with the total parameter count varying slightly based on the head output size and number of tasks we are training on. We will include a comprehensive hyperparameter table in the appendix. In particular, our GCON backbone consists of 16 layers with a hidden dimension of 64 and a dropout rate of 0.3, utilizing ELU as the primary activation function. The architecture employs channels/filters across indices [0], [1], [2], [4], [0,1], [1,2], and [2,4] with sum aggregation and includes 1 Pre-MP and 1 Post-MP layer. Node features are initialized using degree, clustering coefficient, and triangle counts. For optimization, we use AdamW ($\eta=10^{-3}$, $w_d=0$) with a 50-epoch cosine warmup scheduler. Training is conducted with a batch size of 256 for small graphs and 128 for large graphs, using a weighted sum loss strategy where each task weight is set to 1.0. 4) See response to Reviewer iW2r (Q3).
>
> W5) Adapting and running every existing architecture is computationally prohibitive. We selected a prominent SOTA representative to base our analysis on, providing meaningful insights into the specific successes and challenges of the MTL transfer process.
>
> W6) We provide tables for experiments on larger graphs and size extrapolations for the pretraining–finetuning framework from Section 5 (to appear in the appendix; see responses to Reviewers iW2r (Q3) and BXjf (Q3).
>
> W7) We recorded inference time post-pretraining, during finetuning across the 6 CO problems. For a fair comparison, all tasks utilized a single NVIDIA L40S GPU. Pretraining times range from a few minutes (MaxCut, Coloring) to several hours (Backbone, MDS, MaxClique). The table below gives the time in seconds.
>
> | Problem | BA-Large | BA-Small |
> |--------|---------|----------|
> | MaxCut | 6.23 | 2.23 |
> | MaxClique | 48.95 | 32.58 |
> | MDS | 46.42 | 10.65 |
> | MIS | 65.11 | 22.78 |
> | MVC | 137.52 | 31.39 |
> | Coloring | 6.38 | 2.84 |

---

> > ### Author Rebuttal · Reviewer_dqgc · 2026-04-03
> >
> > The rebuttal clarifies the authors’ intended scope, but it does not fully address my main concerns. These key points are hard to address in a short rebuttal.
> >
> > First, the empirical study still remains limited to a narrow set of closely related graph-based CO problems. My concern was not only whether transfer is non-trivial between reducible tasks such as MIS and MVC, but also whether the framework can generalize across genuinely different types of combinatorial optimization problems, such as routing problems like TSP. This point remains unaddressed.
> >
> > Second, the rebuttal continues to center the study around reducible problems, but does not discuss what happens for CO tasks that are not reducible to each other. If the broader goal is to understand transferability across CO problems, then restricting the study to reducible cases significantly limits the scope and practical significance of the conclusions.
> >
> > Third, I remain unconvinced by the novelty claim in W2. The rebuttal states that the contribution lies in integrating different graph neural CO approaches into a novel MTL framework. However, based on the current paper and rebuttal, the method still appears to rely on a single existing backbone, together with an existing loss and known reductions. This is not the same as integrating fundamentally different classes of CO approaches, such as generative methods, reinforcement-learning-based methods, or alternative optimization paradigms. Therefore, the paper still reads primarily as an empirical study built on existing components, rather than a method with substantial technical novelty.

---

> > > ### Author Response · Authors · 2026-04-08
> > >
> > > We thank reviewer dqgc for engaging with our rebuttal. However, we respectfully believe several of the remaining concerns rest on mischaracterizations of our work, and we would like to address them clearly.
> > >
> > > **Narrow set of closely related graph-based CO problems.** We study six problems: MIS, MVC, MaxClique, MaxCut, MDS, and K-coloring. The reviewer repeatedly characterizes these as "closely related," which is not accurate. In fact, only MIS and MVC are trivially complementary (i.e. each other’s complement), and in Sec. 4 they (alongside MaxClique) are used as a preliminary study that builds up to the multi-task setting that considers tasks with substantially non-trivial reductions (despite being linearly/polynomially reducible):
> > > - MDS is related to the set cover problem, which in turn reduces to MIS/MVC in a topology-changing manner
> > > - MaxCut is structurally distinct (partition-based rather than subset-based)
> > > - K-coloring resides in a fundamentally different linear orbit per Filar et al. (2019) and is not linearly reducible to 0-1 integer programming, unlike the other five – it is not any closer to the other tasks than TSP, which the reviewer uses as an example of a different type of CO problem.
> > >
> > > In short, the fact that all six are NP-hard graph problems does not make them "closely related" -- by that logic, virtually all of Karp's 21 problems would be "closely related", which would render a vast majority of reducibility literature trivial. Our leave-one-out results (Table 5) and backbone selection experiments (Table 7) demonstrate meaningfully different transfer behaviors across these problems, precisely because they are (i) not all closely related, and (ii) linear reducibility is not a direct proxy of ease of transferability.
> > >
> > > **Including non-reducible tasks.** Reviewer dqgc asks us to consider CO tasks that are not reducible to each other. Non-reducibility is not only a very strong restriction in itself (again, note that all of Karp’s 21 NP-hard problems are polynomially reducible to each other), but also counters our exact research question, namely whether known reductions are informative for transferability. We thus respectfully argue that a criticism of our focus on reducible tasks misses the paper’s main hypothesis.
> > > However, if we were to interpret this as considering CO tasks that are not linearly reducible to each other, we would like to point out that our paper already addresses this. As aforementioned, K-coloring has no known linear reduction to the other five problems. However, we strikingly demonstrate that despite being the “least reducible” problem in our set, K-coloring shows the largest benefit from multi-task pretraining in our leave-one-out experiments (Table 5).
> > >
> > > **Including TSP and routing problems.** We agree that extending our work to TSP and routing problems is a valid and useful line of future work – previous works have largely only considered transferring across TSP/Hamiltonian cycle or routing problem instances (thus, only staying within their linear reducibility orbit). However, these problems typically operate on complete weighted graphs with fixed node counts and distance-based objectives: a fundamentally different problem structure from the vertex-subset problems we study on unweighted BA/RB graphs of variable size. Including TSP would require an entirely different data pipeline, loss formulation, and decoder, amounting to a different paper. We note that even dedicated multi-task CO papers (GOAL, UniCO, etc.) do not mix vertex-subset and routing problems in a single transfer study. Moreover, we again note that K-coloring is as structurally distant from MIS/MVC as TSP is when measured by the linear orbit framework of Filar et al. (2019), since both K-coloring and TSP/Hamiltonian Cycle occupy different orbits from the remaining problems.
> > >
> > > **Novelty.** The reviewer's post-rebuttal response reiterates that we should test "generative methods, reinforcement-learning-based methods, or alternative optimization paradigms." We respectfully maintain that evaluating every existing CO framework's transferability is far beyond the scope of any single paper: the reviewer effectively asks for a field-wide benchmarking paper that unifies these families of methods under our evaluation framework _in addition to_ our current contributions.
> > >
> > > We thus would like to underline that our contribution is the finding that computational reducibility can guide multi-task pretraining for neural CO: this is a conceptual and empirical contribution, not an architectural or benchmarking one. We are happy to review and revise our work to make our specific contributions clearer, but also believe that we have been transparent about this throughout the paper. We selected GCON as a strong, representative backbone; the insight that reducibility-informed task selection yields efficient transfer is not architecture-dependent and would be expected to hold across frameworks.

---

### Official Review · Reviewer_BnbX · 2026-03-12

**Soundness:** 3
**Presentation:** 2
**Significance:** 2
**Originality:** 2
**Overall Recommendation:** 3
**Confidence:** 2

**Summary:**

The presented paper focuses on solving combinatorial problems using a newly proposed architecture. They first demonstrate the performance on individual tasks, then shift the focus to mutli-task setups to investigate if it is beneficial to pre-train on other tasks. This is particulary done under the aspect that there exist known reductions between the problems.

**Compliance With Llm Reviewing Policy:**

Affirmed.

**Final Justification:**

I keep my recommendation, one of my main concerns is about the unified task vs multi task setup, further it seems that the shift of graph distributions (rather than graph problems) might play a more significant role than currently discussed.

**Key Questions For Authors:**

see questions mentioned above

**Limitations:**

yes

**Strengths And Weaknesses:**

The study of transferability between the presented CO problems is interesting, especially because from a theoretical CS standpoint it is reasonable to expect an interesting connection between reducability and transferability
	- while we know that the problems generally reduce to each other, what is known about reduction wrt to specific distributions? this is arguably more important as the hope is always that the neural CO can pick up on the distributions as in general the problems are NP hard
- One of the main aspects that is considered is efficient transfer of the problems, however, the study is limited to (random) CO datasets on rather small instances without any large instances or size extrapolation experiments
- An alternative way of achieving a CO foundation model and tranferability would be to explicitly only train one one single task (like the SAT mentioning) and then use the known reduction to solve the problems. Is there some inherent advantage of pursuing the multi-task avenue instead?
	- I do not see evidence why multi-loss compared to unified task is more promising. Could it be better to extend the base training set with additional examples (generated from the same base graph, but transformed for a different problem) rather than go the other way of having one set of data and multiple loss functions? A proof of concept experiment in that direction would strengthen the importance of the presented research a lot.
- Overall I feel that the experiments and results in the paper are technically very solid, some settings of interest might be missing (see previous points). My main concern is a bit about the focus of the paper. It presents an abundance of different settings and experiments which makes it hard to pinpoint the most important and interesting result that has a great impact on the research direction.
	- What is the strongest argument for multi-loss training? Especially compared to alternative approaches such as unified (SAT) modeling.

other
- to what extent do you establish a new model? could you be more precise on how it differs from taking GCON and applying it to different problems with (partly?) different loss objectives.
- Is there a benefit in multi-task training for the included tasks? Train on A,B,C is better than just training on A (or followed by an additional ft on A is better than full ft on A). This would also be a valuable argument for the multi-loss setup. I assume Table 7 has some results in that direction but I found it hard to dissect.
- how are the k candidate sets exactly constructed? is it just the first node that differs in the order, or are the sets iteratively predicted?
- in table 3 if you freeze the backbone on the same problem and finetune can you quickly recover the baseline performance?

---

> ### Author Rebuttal · Authors · 2026-03-31
>
> W1) We provide tables for experiments on larger graphs and size extrapolations for the pretraining–finetuning framework from Section 5 (to appear in the appendix; see responses to Reviewers iW2r (Q3+table) and BXjf (Q3+table).
>
> W2 & W3) Examples of unified representation approaches include Learning General Representations Across Graph Combinatorial Optimization Problems [Guo et al.], A Unified Pre-training and Adaptation Framework for Combinatorial Optimization on Graphs [Zeng et al.], and Towards a Generic Representation of Combinatorial Problems for Learning-Based Approaches [Boisvert et al.]. The first is limited to decision problems. The second relies on a MaxSAT/bipartite reduction that is supervised (requiring MaxSAT solver labels) and can suffer from large representation size when applied to complex tasks like k-coloring or TSP. The third notes that even small instances take prohibitive time due to large encodings and are limited to decision problems. We note that reducing all problems to a single task A requires (i) knowing how to reduce each CO problem to A efficiently and (ii) solving A accurately. Our framework instead operates directly on graph structures, avoiding manual reductions. By learning multiple CO problems on graphs, we aim to acquire rich representations that new tasks can reuse (e.g., coverings, partitions). Reductions are often nontrivial; MaxSAT conversions, as in prior work, can be expensive for harder problems (k-coloring, TSP). We also avoid pretraining on generated MaxSAT clauses. GCON architecture has succeeded on CO graphs and avoids common pitfalls of bipartite GNNs for SAT bipartite graphs. For example, Skenderi (A Geometric Perspective on the Difficulties of Learning GNN-based SAT Solvers) shows performance degrades on harder instances because bipartite graphs from k-SAT are negatively curved, creating bottlenecks and thus oversquashing.
>
> W4) Please refer to our rebuttals to reviewers iW2r (W1) and dqgc (W2), where we also address this concern.
>
> W5) Joint training does not yield positive transfer; transfer is usually neutral or slightly negative. However, the goal is to build a strong backbone via carefully selected pretraining tasks (guided by known reductions) so that new tasks can be efficiently finetuned on the go.
>
> W6) The sets are not iteratively predicted, here is how the decoders operate (description added to the appendix): “The candidate sets are constructed by first ranking all vertices in descending order of their model-predicted scores. For each problem type, the algorithm generates K distinct candidate sets (seeds) in parallel to find the best possible solution for that specific graph. In the MaxClique and MIS decoders, the k-th candidate set is initialized with the k-th ranked node, and the algorithm then greedily attempts to add subsequent nodes in the ranked list only if they respect the problem’s constraint, such as forming a clique or an independent set. For the MDS and MVC, the k-th candidate set is constructed by skipping the top k-1 ranked nodes (by setting them to $-\infty$) and then greedily selecting the highest-ranked available nodes until the entire graph is covered. Maxcut simply picks nodes with a score higher than 0.5 to be in the first set and the rest in the second and counts the number of cuts created by this partition, while coloring simply counts the violations (adjacent nodes of the same color).”
>
> W7) Fine-tuning a linear layer on frozen GNN representations converges rapidly: approximately 5 epochs for MIS and 50 for MVC, much faster than fine tuning the full model (approx. 100–150 epochs) or training baselines (approx. 200–300 epochs).

---

> > ### Author Rebuttal · Reviewer_BnbX · 2026-04-04
> >
> > Thank you for the detailed rebuttal.
> >
> > My concern about the unified-task approach still remains, I think a quantitative comparison to highlight the advantages of training on the multi-task vs single task (and hardcoded rerepresentation) is needed to fully justify the presented approach. The argument that such reductions are often not trivial I find a bit limited as it functions also as a fundamental assumption/justification for the multi-task approach.
> >
> > On another note, fine-tuning over the entire dataset for several epochs seems still quite compute heavy, considering few-shot adaptions in other domains. (this is more of a note rather than a major reason for my final assessment).

---

> > > ### Author Response · Authors · 2026-04-08
> > >
> > > We thank reviewer BnbX for their thoughtful follow-up. We appreciate that a head-to-head comparison would be informative, and we agree this is a natural direction for future work. However, we would like to clarify why we believe such a comparison is not a prerequisite to justify our approach. Our goal is not to prove that multi-task pretraining dominates all unified-task formulations, but rather to evaluate whether reducibility can guide task selection in multi-task pretraining to support efficient adaptation without converting every instance into a different representation. Both approaches represent valid pathways toward foundational models for CO, with different trade-offs. As we discuss in the original rebuttal (response to W2 & W3), existing unified-task methods carry significant limitations that we circumvent, which is a contribution in itself:
> > > - Requiring knowing the exact reduction and accordingly building auxiliary graph representations
> > > - Up-to quadratically growing instance sizes when converted to unified (e.g. SAT) representations and prohibitive scaling as a result
> > > - The growing instance sizes exacerbating oversmoothing, oversquashing and underreaching issues that bottleneck the quality of learned graph representations
> > > - Significant graph distribution shifts across SAT instances, thus requiring large amounts of pretraining data across wide graph distributions
> > > - Supervised pretraining (depending on method, e.g. Zeng et al.)
> > >
> > > These points do not necessarily prove that our approach is superior – there may be certain problem transferability settings in which large amounts of pretraining data are available and reductions to unified graph representations are sufficiently scalable so that a unified-task framing is preferable – but we are confident that they are sufficient to justify why our approach is _also_ worth studying.
> > >
> > > We also note that a fair comparison across the aforementioned unified-task methods, which also vary significantly in methodology amongst themselves, require us to reimplement or at the very least adjust these methods to our framework (something we have considered with Zeng et al., but have not succeeded as they do not provide an open-source implementation nor sufficiently clear descriptions of their exact methodology for us to replicate) – a substantial engineering effort that is orthogonal to our main contribution.
> > >
> > > Finally, reviewer BnbX makes a valid point on the non-triviality of reductions that we would like to clarify: In this context, when we say reductions are nontrivial, we essentially refer to the practical challenges of implementing and learning them, rather than an existence of a linear/quasi-linear reduction: e.g., the aforementioned quadratic blow-up of SAT/Max-SAT encodings, or distribution shifts that occur when implementing reductions all affect the quality of the learned representations as well as how efficiently we can learn them in a unified-task setting; challenges that our multi-task method largely avoids.
> > >
> > > We hope that this additional discussion alleviates the reviewer’s concerns; we will clarify these points (particularly the differences in approach and trade-offs re: unified-task vs our multi-task learning) in our revision, in addition to points we addressed in the initial rebuttal (e.g. size extrapolation experiments, clarity re: model formulation and candidate set construction). If satisfied with these clarifications, we hope they will consider raising their score in their final assessment.

---

### Official Review · Reviewer_BXjf · 2026-03-13

**Soundness:** 3
**Presentation:** 3
**Significance:** 2
**Originality:** 2
**Overall Recommendation:** 4
**Confidence:** 4

**Summary:**

The authors take inspiration from algorithmic theory, where the computational complexity of new algorithms can be obtain by reducing them to other known algorithms, and explore whether a similar approach can be used to develop a model that enables transferability and reducibility between different graph CO tasks.
They leverage the GCON network introduced by Wenkel et Al. and extend it by using energy based loss functions (one for each of the six CO tasks that they study).
The authors first perform a study on the pairwise transferability of MIS, MVC and MaxClique, then proceed with a study on multi-task transferability. While the former requires training on a task and then finetuning on the second one, the latter does the same on group of tasks and requires also appropriately swapping the MLP heads for each task.

**Compliance With Llm Reviewing Policy:**

Affirmed.

**Final Justification:**

The authors have addressed most of my concerns and I have increased the score to weak accept. However looking at the other reviews (and the corresponding rebuttal), I feel the paper needs more work.

**Key Questions For Authors:**

- Could you please clarify the results in Table 2? Are they on max clique size on one graph? Please provide additional details on the dataset statistics.
- What are the required times to solve tasks in Table 2?
- Could you please provide the results on scalability? It would be interesting to see experiments on the scalability of graph size.
- Could you please provide results on transferability between MIS (G), MVC (G) and complement(MaxClique(G)) instead of MaxClique(complement (G))?

**Limitations:**

The paper has impact statement

**Strengths And Weaknesses:**

## Strengths
- Implementation: When constructing the k potential solution sets S, k scales better compared to GCON because is parallelized.
- The paper has thorough analysis of the obtained results. Additionally, the paper also provide enough details of the parameters for code reproducibility.
- The paper is well-written and technically sound. The results have cohesive and coherent reasoning and explanation.

## Weaknesses
The problem is interesting. However the results are not significant. Here are a few comments:
- With respect to the claim of transferability and reducibility (line 371): Results (the transferability) seem to be due to the learned graph structure, rather than the model being able to leverage the reducibility. This is substantiated by the fact that if any part of the reduction involves modifying the graph distribution, then transferability doesn't work.
- Significance: Transferability (for instance between MIS and MVC) seems to be due to task equivalence rather than learning on the reduction between the different problems. Additionally, according to results in Table 4, improvements seem to be due mostly to finetuning rather than transferability.
- Originality: The transferability is tested (Table 3) on tasks whose solution are deterministically findable by reformulating the dual/complementary problem, the claims would be much stronger if substantiated by testing on reducing tasks that aren't complimentary.

---

> ### Author Rebuttal · Authors · 2026-03-31
>
> W1) The claim at line 371 states that if a new task has a task in the pretraining set to which it is efficiently reducible, we can transfer efficiently even with a frozen backbone, which would not happen otherwise. Finetuning is able to isolate task A’s information among many pretrained tasks and transfers effectively with a simple MLP head. Section 4 shows that good transfer correlates with simple reductions, while Section 5 extends this to MTL. This suggests MTL benefits from pretraining on a sufficiently diverse set, allowing new tasks to select the most similar pretrained task via computational reducibility, making reducibility a useful guiding principle. Moreover, transferability does work even when we have distribution shifts. Pretraining on MIS and fine tuning with a frozen backbone on MaxClique gives MC of sizes 16.12, which still beats local message passing GNNs and sits just under GFN results (16.24). We observe the impact of distribution shift the most when implementing the actual reduction (MC -> MIS, so $G$ -> $\bar{G}$), and feeding $\bar{G}$ through the pretrained model for MIS and finding the MIS on $\bar{G}$, which corresponds to the MC on $G$. We still get 15.52 in that case using a frozen backbone, which would beat any local message passing GNN baselines, but this example demonstrates the need for fine tuning without freezing the backbone so that the backbone can adapt to distribution shifts and duality gaps.
>
> W2) As we are unsure what the reviewer means by “finetuning rather than transferability”, we assume in our response that when talking about "transferability" here, the reviewer refers to using a frozen backbone as opposed to full fine-tuning. We note that frozen models are already strong: the MIS-pretrained frozen model achieves 16.12 on MaxClique, beating all non-GCON baselines and just behind the second-best (GFN). Frozen models face two inherent limits: duality gaps and distribution shifts from structural modifications, which fine-tuning bridges.
>
> W3) As mentioned at the beginning of section 4, we conjecture that the graph representations learned on MIS should also be sufficient to solve MVC, and vice versa. Furthermore, given the linear relationship between the two, a simple linear layer after message-passing should be able to learn this function using identical representations, and thus we expect the transferred model to quickly recover the original pretrained model performance. Thus section 4.1 serves as a first sanity check on problems where we know a simple reduction. We show that transferability of tasks directly reducible (“equivalent”) to each other isn’t trivial but still works as expected for the most part, due to the duality gap observed because of approximations. We then move on to a simple but more complicated example where the node set remains identical but not the graph structure, and further study the details of transferability for a more complicated reduction. Finally, we extend this to a pretraining-finetuning framework setting in Section 5 where we do not know a priori the reductions between problem, and use the results from Section 4 and [Filar et al. 2019] to choose a backbone that achieves quicker convergence when fine tuning on new tasks compared to training from scratch with a given epoch budget. Results from Figure 2, table 5 and table 7 show strong evidence of transferability for tasks that are not complementary.
>
> Q1) Reported values are average maximum clique (or minimum vertex cover / maximum independent set) sizes over RB-small, containing 6,000 graphs with 200–300 nodes. Dataset details are in the appendix and in the response to review iW2r (Q3).
>
> Q2) Runtimes of Table 2 will be added to the appendix.
>
> Q3) We repeat Section 5.2 experiments by pretraining on small BA graphs and fine tuning on large graphs, and vice versa. We observe strong size extrapolation: for most tasks, pretraining on small then fine tuning on large (and vice versa) performs similarly to training on same-size graphs. Coloring transfers less well: large to small improves over training from scratch but is worse than same-size pretraining, while small to large is similar to training from scratch. We provide below from small to large, large to small will be provided by the camera-ready.
>
> Q4) We are not entirely sure what is being asked here. Is it checking transferability for the different arrow directions in Figure 1 (left)?
>
> | Problem | BA-large Fine-tuned | BA-large Extrapol. | BA-large Diff | BA-large Baseline |
> |--------|---------------------|--------------------|---------------|-------------------|
> | ↑ MaxCut | 2935.8 ± 0.6 | 2936.0 ± 0.1 | +0.01% | 2926.4 ± 2.5 |
> | ↑ MaxClique | 4.31 ± 0.03 | 4.30 ± 0.02 | -0.24% | 4.32 ± 0.01 |
> | ↓ MDS | 109.7 ± 0.2 | 111.9 ± 0.8 | +2.01% | 129.6 ± 12.7 |
> | ↑ MIS | 453.5 ± 0.2 | 453.9 ± 0.2 | +0.08% | 452.6 ± 0.1 |
> | ↓ MVC | 550.5 ± 0.4 | 550.9 ± 0.3 | +0.06% | 554.5 ± 1.2 |
> | ↓ Coloring | 31.4 ± 5.3 | 53.8 ± 12.5 | +71.26% | 53.5 ± 4.5 |

---

> > ### Author Rebuttal · Reviewer_BXjf · 2026-04-03
> >
> > Thank you for the rebuttal. I still think that the core idea of the paper has some  fundamental issues. Mainly the results seem to be linked more to the alignment between the graph distributions with respect to the tasks rather than the model being able to show evidence of reduction or transferability between the tasks themselves. However, the proposed problem is interesting.
> >
> > There are couple of issues that could make the paper stronger: (1) The transferability is being used loosely without much rigorous evidence (for example, it is impacted by graph distribution shift); (2) the running times are missing;
> >
> > Note For Q4: The idea is formulating the task such that it asks for the "complement of the solution to maxclique on G" rather than the "solution of maxclique on the complementary graph of G"
> >
> > I will keep my score.

---

> > > ### Author Response · Authors · 2026-04-07
> > >
> > > We thank reviewer BXjf for their constructive feedback. We would first like to clarify the following point: We agree with the reviewer that computational reducibility alone does not guarantee immediate transferability in neural CO settings; our intended claim is narrower: computational reducibility is a useful prior for selecting pretraining tasks, and its usefulness depends on how much representation/distribution shift the reduction induces. We evaluate our claims in two regimes in Sec. 4:
> > > - In the former, we show that when the reduction preserves graph structure (MIS <-> MVC), even a frozen-backbone transfer is very strong and rapidly converging – while we understand the reviewer’s original concern that “reducing tasks that aren't complementary” is more original/interesting, this first study serves as a useful sanity check of our proposed setup, and is valuable in being the first study (to our knowledge) to concretely demonstrate these reductions in neural CO as well as within the narrative of the paper to segue into the multi-task learning setup.
> > > - In the latter study, we show that when the reduction changes topology (MaxClique($G$) = MIS($\bar{G}$)), pretraining still provides a strong initialization (which is competitive against other neural CO benchmarks), but backbone fine-tuning becomes important to bridge the induced structural shift.
> > >
> > > In light of these findings, we would argue that demonstrating this relationship between graph distribution shifts and reducibility is a contribution of the paper rather than a weakness: Sec. 4 essentially demonstrates when reducibility directly translates to transferability and when it doesn’t. Specifically, we see that the more topology-preserving a reduction is, the easier to transfer across tasks. We then demonstrate that fine-tuning rapidly recovers original performance precisely because closing this distribution gap is easier than training from scratch. We hope this clarifies our point of view: These results are better interpreted as evidence that reducibility is a guiding principle rather than a sufficient condition. We will integrate these points into our discussion of Sec. 4 in our revision.
> > >
> > > In the meantime, we note that the size extrapolation study we provided in our rebuttal shows considerable robustness across graph sizes: BA-small graphs are 200-300 nodes each, while BA-large graphs consist of 800-1200 nodes. Graph size is of course one of several axes to consider (e.g., this study can be extended to different graph generation paradigms or densities etc.), but it demonstrates a degree of robustness of the learned representations to certain distribution shifts.
> > >
> > > **Use of the term "transferability":** We understand the reviewer’s concern, as the term “transferability” is used contextually to refer to both frozen-backbone and full fine-tuning setups, both of course established uses of transfer learning. We will revise the wording appropriately to make this distinction explicit throughout the paper, and we encourage the reviewer to point us to any specific phrasing of transferability that they think requires a revision.
> > >
> > > **Inference times for Table 2** over the test set (500 graphs, NVIDIA Tesla V100) are provided below; we will update the original table accordingly (filling in any missing results). Note that despite comparable times to the other methods on the RB-small dataset, GFN & Gurobi scale substantially worse compared to the GNN-based methods on larger graphs. MVC times for Gurobi & GFN are not listed, but would be approximately equal to MIS as one would solve for MIS and invert the solution.
> > >
> > > |Method|Type| MVC ↓| MVC Time | MClique ↑| MClique Time | MIS ↑| MIS Time |
> > > |-|-|-|-|-|-|-|-|
> > > | True Size | — | 206.95 | — | 19.07 | — | 20.07 | — |
> > > | Gurobi | OR | — | — | 19.05† | 1:55 | 19.98†  | 47:34 |
> > > | GFN | SSL | — | — | _16.24_ | 0:42 | **19.18** | 0:32 |
> > > | GCN | SSL-GNN | 221.56 ± 0.05 | 1:29 | 15.33 ± 0.04 | 0:57 | 17.67 ± 0.20  | 0:45 |
> > > | GIN | SSL-GNN | 221.63 ± 0.23 | 1:24 | 15.28 ± 0.14 | 0:55 | 17.50 ± 0.04 | 0:35 |
> > > | GATv2 | SSL-GNN | _220.76 ± 2.26_ | 1:26 | 15.51 ± 0.03 | 1:02 | 17.58 ± 0.07 | 0:57 |
> > > | GCON | SSL-GNN | **211.69 ± 0.16** | 1:46 | **16.92 ± 0.13** | 1:20 | _18.12 ± 0.11_ | 1:05 |
> > >
> > > **Q4:** Thank you for the clarification. By Lemma B.1, $V$ \ MaxClique($G$) = MVC($\bar{G}$), which shifts the complement to the output. There is no known reduction linking MaxClique($G$) to MIS($G$) or MVC($G$) on the same graph without the complement – so transferring to MIS/MVC($G$) (considering that $G$ and not $\bar{G}$ is our graph of interest) always requires reducing from the MaxClique of the complement – which is why we consider transferring only from MaxClique($\bar{G}$).
> > >
> > > We again thank the reviewer, and hope that this additional discussion further alleviates their concerns; if satisfied with our response, we hope they will consider a score increase.

---

### Official Review · Reviewer_iW2r · 2026-03-13

**Soundness:** 3
**Presentation:** 2
**Significance:** 3
**Originality:** 3
**Overall Recommendation:** 3
**Confidence:** 5

**Summary:**

This work studies the transferability of neural solvers for combinatorial optimization problems on graphs, with a focus on building unified models capable of handling multiple tasks. The authors propose a framework combining the Graph Combinatorial Optimization Network (GCON) with energy-based unsupervised loss functions, sequential decoders, and multi-task pretraining and fine-tuning strategies. They systematically evaluate task-pair transfer (MIS, MVC, MaxClique) and multi-task transfer (MaxCut, MDS, K-coloring, etc.), leveraging known polynomial reductions to guide task selection and backbone pretraining. Experiments demonstrate that pretraining on carefully selected tasks accelerates fine-tuning, often achieving performance comparable to fully trained single-task models while requiring fewer epochs.

**Compliance With Llm Reviewing Policy:**

Affirmed.

**Key Questions For Authors:**

Q1: Could you clarify the implementation details of the GCON multi-scale wavelet filters and their specific contribution to transferability compared to GNN layers?

Q2: In the transfer experiments, how do you ensure the feasibility of the obtained solutions with respect to the problem constraints? Although a sequential decoder is mentioned, it would be helpful if the authors could clarify how constraint satisfaction is guaranteed during transfer and whether infeasible solutions may still occur in practice.

Q3: The experiments are mainly conducted on relatively small-scale instances, what is the performance of the proposed approach on larger-scale graphs? It would also be interesting to evaluate the proposed approach on well-established benchmark instances, such as the DIMACS instances, which are commonly used for MaxClique, MIS, etc.

**Limitations:**

Yes

**Strengths And Weaknesses:**

Strengths:
The strengths of this work include its novel combination of theoretical polynomial reductions with neural combinatorial optimization, a comprehensive experimental evaluation covering ablation studies, pairwise, and multi-task transfer, and an effective methodology that integrates GCON embeddings, energy-based loss functions, sequential decoders, and multi-task pretraining. It convincingly demonstrates task transferability, showing that carefully selected pretraining tasks can accelerate fine-tuning, reduce computational resources, and achieve performance comparable to fully trained single-task models.

Weaknesses:
While the proposed framework is carefully designed and empirically effective, many of its core components, such as the GCON encoder, energy-based loss functions, and sequential decoders, are largely adaptations or extensions of existing methods. Furthermore, the selected tasks in the study (MIS, MVC, MaxClique, MaxCut, MDS, K-coloring) are all closely related graph combinatorial optimization problems. However, there exist many other types of graph CO problems, such as TSP or vehicle routing, it is unclear whether the transferability and pretraining strategies can generalize to a broader class of graph CO problems.

---

> ### Author Rebuttal · Authors · 2026-03-31
>
> W1 The different components are not necessarily novel, but we do not claim to propose a new method per se, but rather study how we can leverage known strategies for single task learning of graph CO problems and extend it to a multitask learning / transfer learning setting where we can avoid training a new model from scratch for every new CO problems we want to solve (we still argue that there is novelty in our framework though, see reviewer dqgc, W2). We suggest and show empirical evidence that computational reducibility is a useful starting point to guide this framework.
>
> W2 Moreover, while it is true that we do not include TSP, which is a very popular CO problem, we believe we selected a diverse and large enough set of CO problems, with K-coloring being significantly different then the others (as much as TSP is, considering the linear orbits described in Linearly-growing Reductions of Karp's 21 NP-complete Problems [Filar et al.]). Moreover, the current setting does not have a natural way to learn TSP, since we are working on BA and RB graphs which a) have different sizes (num of nodes) and b) do not have distances. Even if we limit ourselves to HCP (which does not require distances) instead of TSP, we would be limited to working on graphs of the same size (since HCP would require an NxN output, thus N needs to be fixed) similar to what is done in Unsupervised Learning for Solving the Travelling Salesman Problem [Yimeng et al.] with the datasets TSP50, etc. These TSP datasets would not be interesting in our case since they give cliques and are specific to TSP.
>
> Q1 The GCON multi-scale wavelet filters are as described in [Wenkel et al.], that is aggregation operations of the form $F_k(X) := m(P^{k} X)$ and comparison operations of form $F_{2^{j-1}, 2^j}(X) := m((P^{2^{j-1}} - P^{2^j})X)$. In practice, we use the filters $F_0, F_1, F_2, F_4$ and $F_{0,1}, F_{1,2}, F_{2,4}$ and an attention mechanism lets the model learn which filters to leverage to extract an expressive representation. Classical GNN architectures such as GCN perform well when we can treat smoothness as an inductive bias, which does not apply to CO problems, hence the use of GCON. In fact, our ablation study shows that GCON modules can learn a representation that allows us to learn accurate approximations of MIS, MVC and MaxClique (and MaxCut, MDS in [Wenkel et al.]), significantly better than GCN, GIN and GAT. Traditional GNNs would therefore be unable to be used for effective transfer if they cannot learn an expressive enough representation in the first place.
>
> Q2 Infeasible solutions cannot happen in practice due to the nature of the decoders, since they iteratively add vertices to the solution set only if they satisfy the CO problem constraints. When we do transfer learning (from Task A to B), we use the learned weights of Task A as the initialization of the backbone with a new randomly initialized MLP head, and optimize the loss function of Task B, before feeding the output of the model to the decoder of Task B. Therefore we are guaranteed a feasible solution to Task B. More specifically, we added a description of the decoders in the appendix, see response to Reviewer BnbX (W6) for more details.
>
> Q3) We provided results on BA and RB small datasets, which include 6000 graphs of sizes 200-300 nodes. We added the following datasets description in the appendix : “We conduct our experiments on the same synthetic datasets used in GCON, which are commonly used in the field of neural graph combinatorial optimization. In all experiments, we use datasets of size 6000 with 200-300 nodes (small) and 800-1200 nodes (large). In Section 4, we use graphs generated from the RB model. For RB graphs, we can specify the number of cliques (n) and the number of nodes per cliques (k). We use $n \in [20,25]$ and $k \in [5,12]$ for RB-small and $n \in [40,55]$ and $k=20$ for RB-large. We then use Barabási–Albert (BA) graphs in Section 5. For BA graphs, we can specify how many edges are attached from each new node, which we set to 4 for both BA-small and BA-large.” Running our pretraining-finetuning setup on BA-large, we get that pretraining on MDS-MIS-Coloring on the large dataset performs as well as on the small datasets. Additional results also show that pretraining on small graphs and fine tuning large graphs give almost the same results as if we would pretrain on large graphs (except for K-coloring) (see Review BXjf Q3 and table). Finally, we don’t think that the DIMACS dataset is suitable for our setting, since it is not a dataset we can easily train-test and is specific to certain CO problems only.
>
> | Problem | Fine-tuned | Baseline |
> |--------|-----------|----------|
> | ↑ MaxCut | 2935.82 ± 0.63 | 2926.39 ± 2.46 |
> | ↑ MaxClique | 4.31 ± 0.03 | 4.32 ± 0.01 |
> | ↓ MDS | 109.72 ± 0.16 | 129.64 ± 12.74 |
> | ↑ MIS | 453.51 ± 0.17 | 452.59 ± 0.07 |
> | ↓ MVC | 550.55 ± 0.36 | 554.53 ± 1.22 |
> | ↓ Coloring | 31.40 ± 5.29 | 53.48 ± 4.53 |

---

> > ### Author Rebuttal · Reviewer_iW2r · 2026-04-03
> >
> > Thank you for the rebuttal. The authors have addressed most of my concerns. I still believe that the trained model should be able to generalize to DIMACS benchmark instances, as they are widely regarded as challenging and representative benchmarks.
> >
> > My score remains unchanged.

---

> > > ### Author Response · Authors · 2026-04-08
> > >
> > > We thank the reviewer for the positive assessment and for acknowledging that most concerns were addressed.
> > >
> > > We appreciate the suggestion and understand DIMACS is a standard reference point for MaxClique/MIS. However, as they are likely aware, DIMACS is a collection of individual fixed graphs (typically one or few per instance), not a dataset with train/test splits. Our framework studies pretraining and fine-tuning across tasks, which requires batched training over graph distributions — something DIMACS is not designed for. One cannot meaningfully pretrain on 6,000 BA graphs and then "fine-tune" on a single DIMACS instance; our experimental framework therefore does not translate beyond fully-frozen, zero-shot transfer.
> > >
> > > That said, we are happy to offer two things in the revision:
> > > 1. We can report inference-only, zero-shot transfer results by running our pretrained models on select DIMACS instances (treating them as a test set) to demonstrate that the learned representations produce reasonable solutions on out-of-distribution real-world graphs -- of which we provide a preliminary version below.
> > > 2. We can include a discussion of how our framework could be adapted to instance-specific settings in future work.
> > >
> > > We want to be transparent, however, that DIMACS performance is not the contribution of this paper — our focus is on understanding whether reducibility can guide multi-task transfer, and the synthetic graph distributions we use are standard in the neural CO literature.
> > >
> > > To conduct inference on DIMACS instances as proposed above, we use the evaluation framework from _FrontierCO: real-world and large-scale evaluation of machine learning solvers for combinatorial optimization_ (Feng et al.) by pretraining our model on RB graphs (due to time constraints, we train on RB-small for the time being, and promise an updated version trained on RB-large graphs following Feng et al. by the camera-ready) and testing on DIMACS instances. We also report results based on a model with randomly initialized weights for reference. We do this for the MIS task by finding the MIS of the complement of DIMACS MaxClique graphs, which is equivalent to the MaxClique of the original DIMACS graphs for which the best known sizes are available (2nd DIMACS Challenge, [Johnson & Trick 1996]). We observed that our pretrained model comfortably outperforms random initialization as expected on almost all graphs, and obtains reasonable zero-shot results on many instances. It should however be noted that it isn’t surprising that we do not attain near-SOTA results in the zero-shot setting due to (i) considerable distribution shift, and (ii) the fact that as Feng et al. notes, ML-based solvers typically lag behind SOTA solvers on these hard instances and distribution shifts in general.
> > >
> > > We additionally promise to expand on the results below by adding results for MDS following the same framework in our revision, and replicating the performance gap study in Feng et al. for MIS & MDS under fair training & evaluation settings. We thus hope the reviewer finds that the preliminary DIMACS study below along with our large-graph experiments (pretraining on BA-small & finetuning on BA-large, and vice versa), the timing results, and the dataset descriptions provided in the rebuttal sufficiently address the remaining concerns, and we hope they consider increasing their score accordingly.
> > >
> > > | Graph name | Pretrained | Random | Best known |
> > > |-|-:|-:|-:|
> > > | C1000.9 | 56 | 29     | 68         |
> > > | C125.9 | 32 | 25     | 34         |
> > > | C2000.5 | 11 | 9      | 16         |
> > > | C2000.9 | 48 | 28     | 80         |
> > > | C250.9 | 36 | 28     | 44         |
> > > | C400.5 | 1  | 8      | 18         |
> > > | C500.9 | 50 | 30     | 57         |
> > > | DSJC1000.5 | 11 | 8      | 15         |
> > > | DSJC500.5 | 10 | 8      | 13         |
> > > | MANN_a27 | 117 | 85     | 126        |
> > > | MANN_a45 | 276 | 71     | 345        |
> > > | MANN_a81 | 300 | 122    | 1100       |
> > > | brock200_2 | 10  | 8      | 12         |
> > > | brock200_4 | 13 | 11     | 17         |
> > > | brock400_2 | 21 | 16     | 29         |
> > > | brock400_4 | 23 | 15     | 33         |
> > > | brock800_2 | 16 | 12     | 24         |
> > > | brock800_4 | 16 | 12     | 26         |
> > > | gen200_p0.9_44 | 37 | 28     | 44         |
> > > | gen200_p0.9_55 | 38 | 30     | 55         |
> > > | gen400_p0.9_55 | 48 | 30     | 55         |
> > > | gen400_p0.9_65 | 44 | 31     | 65         |
> > > | gen400_p0.9_75 | 45 | 33     | 75         |
> > > | hamming10-4 | 27 | 16     | 40         |
> > > | hamming8-4 | 16 | 16     | 16         |
> > > | keller4 | 9 | 8      | 11         |
> > > | keller5 | 18 | 15     | 27         |
> > > | keller6 | 37 | 28     | 59         |
> > > | p_hat1500-1 | 10 | 3      | 12        |
> > > | p_hat1500-2 | 52 | 4      | 65        |
> > > | p_hat1500-3 | 79 | 10     | 94         |
> > > | p_hat300-1 | 7 | 5      | 8          |
> > > | p_hat300-2 | 24 | 10     | 25         |
> > > | p_hat300-3 | 31  | 17     | 36         |
> > > | p_hat700-1 | 8 | 3      | 11         |
> > > | p_hat700-2 | 38 | 5      | 44         |
> > > | p_hat700-3 | 57 | 10     | 62         |

---

### Decision · Program_Chairs · 2026-04-30

**Decision:**

Accept (regular)

**Comment:**

This paper explores an interesting direction in neural graph combinatorial optimization – transfer across tasks. Proposed method seems quite heavy methodologically, and reviewers recognize that; however I believe that most of reviewer's concerns do not substantially the originality & significance aspects of the work. Given the additional results provided during the rebuttal phase, I believe that the paper meets the bar for ICML and the community would benefit from discussing and building on top of this work.